# `BIOT`: Biosignal Transformer for Cross-data Learning in the Wild

**Chaoqi Yang[1], M. Brandon Westover[2,3], Jimeng Sun[1]**
[1]University of Illinois Urbana-Champaign, [2]Harvard Medical School
[3]Beth Israel Deaconess Medical Center
{chaoqiy2}@illinois.edu

## Abstract

Biological signals, such as electroencephalograms (EEG), play a crucial role in numerous clinical applications, exhibiting diverse data formats and quality profiles. Current deep learning models for biosignals (based on CNN, RNN, and Transformers) are typically specialized for specific datasets and clinical settings, limiting their broader applicability. This paper explores the development of **a flexible biosignal encoder architecture** that can enable pre-training on multiple datasets and fine-tuned on downstream biosignal tasks with different formats.

To overcome the unique challenges associated with biosignals of various formats, such as mismatched channels, variable sample lengths, and prevalent missing values, we propose Biosignal Transformer (`BIOT`). The proposed `BIOT` model can enable cross-data learning with mismatched channels, variable lengths, and missing values by tokenizing different biosignals into unified "sentences" structure. Specifically, we tokenize each channel separately into fixed-length segments containing local signal features and then re-arrange the segments to form a long "sentence". Channel embeddings and *relative* position embeddings are added to each segment (viewed as "token") to preserve spatio-temporal features.

The `BIOT` model is versatile and applicable to various biosignal learning settings across different datasets, including joint pre-training for larger models. Comprehensive evaluations on EEG, electrocardiogram (ECG), and human activity sensory signals demonstrate that `BIOT` outperforms robust baselines in common settings and facilitates learning across multiple datasets with different formats. Using CHB-MIT seizure detection task as an example, our vanilla `BIOT` model shows 3% improvement over baselines in balanced accuracy, and the pre-trained `BIOT` models (optimized from other data sources) can further bring up to 4% improvements. Our repository is public at https://github.com/ycq091044/BIOT.

## 1 Introduction

Biosignals, such as EEG and ECG, are multi-channel time series recorded at high sampling rates (e.g., 256Hz) in various healthcare domains, including sleep medicine, neurological and cardiovascular disease detection, and activity monitoring. Deep learning (DL) models have demonstrated impressive success in automating biosignal analysis across diverse applications (Yang et al., 2021), encompassing sleep stage classification (Biswal et al., 2018; Yang et al., 2021; Phan and Mikkelsen, 2022), emotion analysis via EEG (Zhang et al., 2020; Suhaimi et al., 2020), action and motor imagery recognition (Venkatachalam et al., 2020), acute stress detection through electrodermal activity (Greco et al., 2021), EEG-based seizure epilepsy classification (Yang et al., 2023; Jing et al., 2023), and ECG-driven cardiac arrhythmia detection (Isin and Ozdalili, 2017; Parvaneh et al., 2019).

Various deep learning methods have been developed for biosignal learning. Some works use 1D convolutional neural networks (CNN) on raw signals (Jing et al., 2023; Nagabushanam et al., 2020;

Dar et al., 2020), while others preprocess the data with short-time Fourier transform (STFT) and employ 2D CNN models on the resulting spectrogram (Yang et al., 2022; Kim et al., 2020; Cui et al., 2020). Researchers also segment the signal and use a CNN segment encoder with a downstream sequence model (Zhang et al., 2019; Biswal et al., 2018; Jing et al., 2020; Almutairi et al., 2021), such as Transformer or recurrent neural networks (RNN), to capture temporal dynamics. Other approaches involve ensemble learning, feature fusion from multiple encoders (Li et al., 2022), and multi-level transformers to encode spatial and temporal features across and within different channels (Lawhern et al., 2018; Song et al., 2021; Liu et al., 2021).

These models (Jing et al., 2023; Yang et al., 2021; Biswal et al., 2018; Kostas et al., 2021; Du et al., 2022; Zhang et al., 2022) predominantly focus on biosignal samples with fixed formats for specific tasks, while real-world data may exhibit mismatched channels, variable lengths, and missing values. In this paper, our objective is to devise a flexible training strategy that can handle diverse biosignal datasets with varying channels, lengths, and levels of missingness. For example, is it possible to transfer knowledge from abnormal EEG detection (a binary classification task with samples having 64 channels, 5-second duration, recorded at 256Hz) to improve another EEG task, such as seizure type classification (a multi-class task with 16 channels and a 10-second duration at 200Hz)? In reality, such data mismatches often arise from varying devices, system errors, and recording limitations. Additionally, it is also important to explore the potential of utilizing various unlabeled data sources.

To apply existing deep learning models to such settings of different biosignals, significant data processing is required to align the formats across multiple datasets. This may involve truncating or padding signals for consistent lengths (Zhang et al., 2022), and imputing missing channels or segments (Bahador et al., 2021). Such practices, however, may introduce unnecessary noise and shift data distributions, leading to poor generalization performance. Developing a flexible and unified model that accommodates biosignals with diverse formats can be advantageous.

In our paper, we develop the biosignal transformer (`BIOT`, summarized in Figure 1), which, to the best of our knowledge, is the first multi-channel time series learning model that can handle biosignals of various formats. Our motivation stems from the vision transformer (ViT) (Dosovitskiy et al., 2020) and the audio spectrogram transformer (AST) (Gong et al., 2021). The ViT splits the image into a "sentence" of patches for image representation while the AST splits the audio spectrogram into a "sentence" for 1D audio representation. We are inspired by that these "sentence" structures combined with Transformer (Vaswani et al., 2017) can handle variable-sized inputs.

However, it is non-trivial to transform diverse biosignals of various formats into unified "sentence" structures. This paper proposes `BIOT` to solve the challenge by a novel **biosignal tokenization module** that segments each channel separately into tokens and then flattens the tokens to form consistent biosignal "sentences" (illustrated in Figure 2). With the design, our `BIOT` can enable the knowledge transfer cross different data in the wild and allow joint (pre-)training on multiple biosignal data sources seamlessly. Our contributions are listed below.

- **Biosignal transformer** (`BIOT`). This paper proposes a generic biosignal learning model `BIOT` by tokenizing biosignals of various formats into unified "sentences."

- **Knowledge transfer across different data.** Our `BIOT` can enable joint (pre-)training and knowledge transfer across different biosignal datasets in the wild, which could inspire the research of large foundation models for biosignals.

- **Strong empirical performance.** We evaluate our `BIOT` on several unsupervised and supervised EEG, ECG, and human sensory datasets. Results show that `BIOT` outperforms baseline models and can utilize the models pre-trained on similar data of other formats to benefit the current task.

## 2 `BIOT`: Biosignal Transformer

As shown in Figure 1, our `BIOT` encoder cascades two modules: (i) the **biosignal tokenization** module that tokenizes an arbitrary biosignal (variable lengths, different channels, and missing values) into a "sentence" structure. This design can potentially enable previous language modeling techniques (Devlin et al., 2018; Liu et al., 2019; OpenAI, 2023) to empower the current biosignal models; (ii) a **linear transformer** module that captures complex token interactions within the "sentence" while maintaining linear complexity. After that, we also discuss the application of `BIOT` in different real-world settings. For all the notations used in the paper, we summarize them in Appendix A.

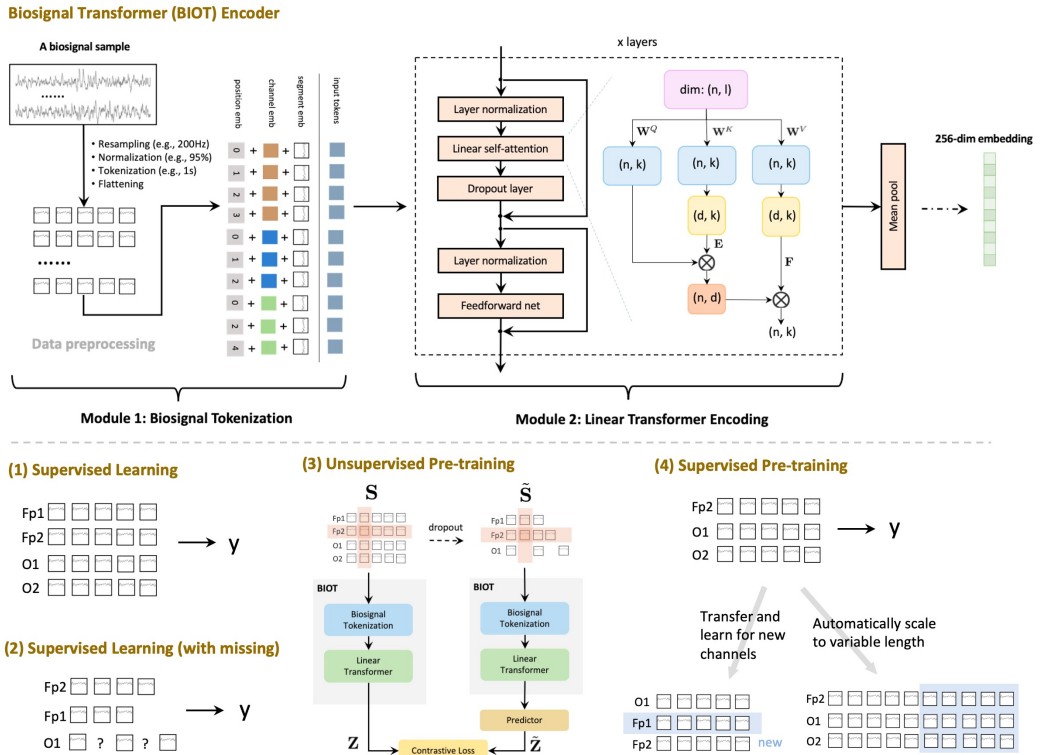

Figure 1: Biosignal Transformer (`BIOT`). (Upper) Given a new data sample, we initially perform data preprocessing (resampling, normalization, tokenization, and flattening) to create a biosignal "sentence" using the **biosignal tokenization module**. We then learn the complex interactions within the "sentence" through the **linear transformer module**. (Lower) `BIOT` encoder is versatile, enabling supervised learning on complete data, data with missing values, and pre-training and fine-tuning across diverse data formats and tasks.

## 2.1 Module 1: Biosignal Tokenization

**Motivation.** The goal of this paper is to model heterogeneous biosignals (e.g., EEG samples with different channels for different tasks) with a unified model. For example, common EEG samples (Lopez et al., 2015), such as those for seizure detection, are recorded at 256Hz in the international 10-20 system [1] for 10-second long (Klem et al., 1999). With standard 16 montage editing, the sample is essentially a multi-channel time-series, represented as a matrix of size (16, 2560). However, format mismatch may prevent the applications on other similar data, such as **different sampling rate** (e.g., 200Hz vs 256Hz) (Jing et al., 2023), **mismatched channels** (i.e., different datasets have their own novel channels), **variable recording duration** (i.e., 30s per sample vs. 10s) (Zhang et al., 2022), **missing segments** (i.e., part of the recording is damaged due to device error). Thus, existing models may fail to utilize the mismatched data from different datasets.

Our `BIOT` solves the above challenges of mismatches by the following steps. We assume one biosignal sample as $\mathbf{S} \in \mathbb{R}^{I \times J}$. Here, $I$ as the channels and $J = duration(s) \times rate(Hz)$ as the length.

- **Resampling.** We first resample all data to the same rate, denoted by $r \in \mathbb{R}^+$ (such as 200Hz), by linear interpolation. The common rate $r$ could be selected following clinical knowledge of a certain biosignal. For example, the highest frequency of interest in both EEG and ECG signals is commonly around 100 Hz, and thus 200 Hz or 250 Hz can be suitable for typical EEG or ECG applications, according to Nyquist-Shannon sampling theorem (Nyquist, 1928; Shannon, 1949).

- **Normalization**: To alleviate the unit difference and amplitude mismatch across different channels and datasets, we use the 95-percentile of the absolute amplitude to normalize each channel. Formally, each channel $\mathbf{S}[i]$ is normalized by $\frac{\mathbf{S}[i]}{\text{percentile}([|\mathbf{S}[i,1]|,|\mathbf{S}[i,2]|,...,|\mathbf{S}[i,J]|], \, 95\%)}$.

---

[1]https://en.wikipedia.org/wiki/10-20_system_(EEG)

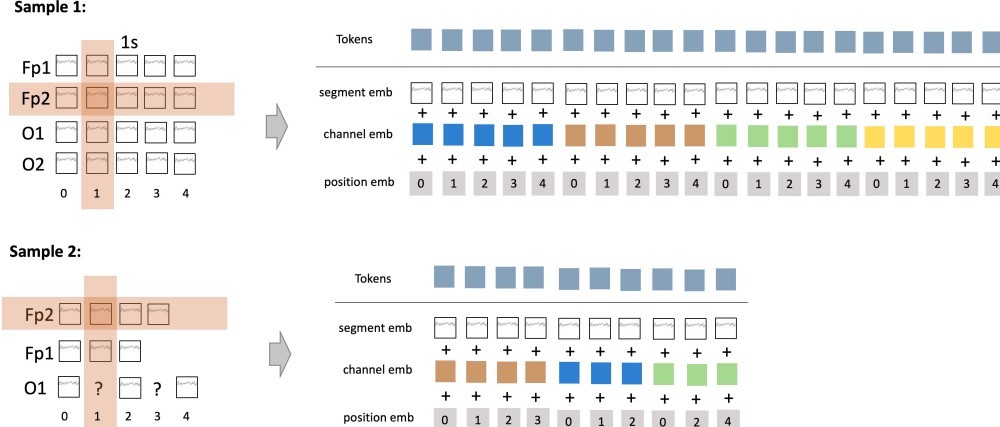

Figure 2: Biosignal Tokenization (no segment overlap in the examples). **Sample 1** has four channels (Fp1, Fp2, O1, and O2) for 5 seconds. We tokenize each channel into segments/tokens and then parameterize these 20 segments with three embeddings. On the right, we use different colors to represent the channels (blue-Fp1, brown-Fp2, green-O1, and yellow-O2). **Sample 2** has mismatched channels (no O2), variable lengths (Fp1 and Fp2 are shorter and flipped), and missing values (in O1). Using our method, we can still tokenize Sample 2 in a comparable "sentence". On both samples, the peach orange area indicates the spatial- or temporal-relevant tokens w.r.t. the highlighted token.

- **Tokenization**: For handling length variation, we tokenize the recording of each channel into segments/tokens of length $t \in \mathbb{R}^+$, and neighboring tokens can have overlaps of length $p \in \mathbb{R}$ ($p < t$, e.g., $t = r$ and $p = 0.5r$ corresponds to 1s and 0.5s). Thus, the $k$-th token ($k = 1, 2, .., K$) in the $i$-th channel can be represented by slicing $\mathbf{S}[i, (t - p)(k - 1) : (t - p)(k - 1) + t]$. The number of tokens $K$ per channel is limited by the inequality: $(t - p)(K - 1) + t \leq J$. Here, the overlap $p$ is essential to maintain the temporal information for shorter signals. For example, if the length of a signal is 3 seconds (i.e., $J = 3r$), the set $t = r, p = 0$ only generates 3 tokens per channel, while the set $t = r, p = 0.5r$ gives 5 tokens per channel (see Appendix C.3). In cases of missing values, we drop the corresponding tokens directly (as shown in **Sample 2**). Note that, our tokenization applies to each channel, separately, which is different from previous works (Almutairi et al., 2021; Du et al., 2022) that split all channels together (which cannot work on **Sample 2**).

- **Flattening**: We finally flatten tokens from all channels into a consistent "sentence".

The above steps are non-parametric. To study the effect of sampling rate $r$, token length $t$, and overlap $p$, we provide ablation studies and insights in Appendix C.3. In the following, we design the token embedding for the biosignal "sentence", which combines information from three aspects.

- **Segment embedding.** We leverage fast Fourier transform (FFT) to extract an energy vector for each token where every dimension indicates the energy of a specific frequency. A fully connected network (FCN) is then applied on the energy vector to transform it into the segment embedding. Directly modeling the raw time-series to learn a segment embedding can be another option.

- **Channel embedding (spatial).** We learn an embedding table for all different channels and add the corresponding channel embedding to the token. Each color represents one channel in Figure 2.

- **Positional embedding (temporal).** In biosignals, the token order within the channel captures temporal information. We thus add *relative* positional embedding to the token embedding by using the sinusoidal/cosine functions (Vaswani et al., 2017), which does not need learnable parameters.

The above three embeddings are summed to create the token embedding, which can effectively capture the segment features as well as the spatio-temporal features. Finally, we denote the tokenzied "sentence" as $\mathbf{X} \in \mathbb{R}^{N \times l_1}$ where $N$ is the number of tokens and $l_1$ is the embedding dimension.

## 2.2 Module 2: Linear Transformer

**Transformer with linear complexity for long biosignal "sentence".** Next, we want to leverage the Transformer model (Vaswani et al., 2017) for modeling the "biosignal sentence". However, our transformation may lead to long "sentences" due to multiple channels. For example, the "sentence"

of a 64-channel EEG signal for 20 seconds (without overlaps) may have $64 \times 20 = 1280$ tokens, and longer with the overlaps. Given that the original Transformer model is known to have quadratic complexity in both time and space, we adopt the linear attention mechanism (Wang et al., 2020; Katharopoulos et al., 2020) for biosignal learning applications.

Formally, let us assume $\mathbf{W}^K, \mathbf{W}^V, \mathbf{W}^Q \in \mathbb{R}^{l_1 \times l_2}$ be the key, value, and query matrices. Our self-attention module uses a rank-$d$ approximation for the softmax attention ($N \times N$) by reduced-rank parameter matrices $\mathbf{E}^\top \in \mathbb{R}^{N \times d}$, $\mathbf{F} \in \mathbb{R}^{d \times N}$ (where $d \ll N$). The output $\mathbf{H} \in \mathbb{R}^{N \times l_2}$ is,

$$\mathbf{H} = \text{Attention}(\mathbf{XW}^Q, \mathbf{EXW}^K, \mathbf{FXW}^V) \tag{1}$$

$$= \underbrace{\text{softmax}\left(\frac{(\mathbf{XW}^Q)(\mathbf{EXW}^K)^\top}{\sqrt{l_2}}\right)}_{N \times d} \cdot \underbrace{\mathbf{FXW}^V}_{d \times l_2}. \tag{2}$$

Main components of this module include one linear self-attention layer and one fully connected network. To enable stable training, we add layer normalization (Ba et al., 2016) before each component, residual connection (He et al., 2016) after each component, and dropout (Srivastava et al., 2014) right after the self-attention (see Figure 1), which improves the convergence.

`BIOT` **Encoder.** An illustration of our proposed `BIOT` encoder is shown in Figure 1 (upper), which comprises the **biosignal tokenization** module and multiple blocks of **linear transformer** modules. We obtain the final biosignal "sentence" embedding by a mean pooling step over all tokens. Note that appending a classification [CLS] token at the beginning of the "sentence" (after Module 1) is also a common option. However, we find it yields a slightly worse performance in our application, and thus we use mean pooling in the experiments.

### 2.3 Biosignal Learning in the Wild

Our proposed `BIOT` encoder can be applied in various real-world biosignal applications (illustrated in the lower part of Figure 1). These applications include (1) standard supervised learning, (2) learning with missing channels or segments, (3) & (4) pre-training on one or more datasets, and fine-tuning on other similar datasets with different input formats.

**(1) Supervised Learning** is the most common setting in the previous literature (Jing et al., 2023; Biswal et al., 2018). With the `BIOT` encoder model, we finally apply an exponential linear unit (ELU) activation (Clevert et al., 2015) and a linear layer for classification tasks.

**(2) Supervised Learning (with missing).** Many real biosignal data have mismatched channels, missing segments, and variable lengths, which prevents the applications of existing models (Jing et al., 2023; Song et al., 2021). Flexible as our model is, `BIOT` can be applied in this setting using the same model structure as in **(1)**.

**(3) Unsupervised Pre-training.** We can jointly pre-train a general-purpose `BIOT` encoder on multiple large unlabeled datasets. In the experiments, we pre-train an unsupervised encoder using 5 million resting EEG samples (16 channels, 10s, 200Hz) and 5 million sleep EEG samples (2 channels, 30s, 125Hz), which is later utilized to improve various downstream tasks.

For the unsupervised pre-training, we take the following steps (a diagram is shown in Figure 1).

- Assume $\mathbf{S}$ is the original biosignal. We first randomly dropout part of its channels and dropout part of the tokens from the remaining channels, resulting in a perturbed signal $\tilde{\mathbf{S}}$.

- We then obtain the embeddings of $\mathbf{S}$ and $\tilde{\mathbf{S}}$ by the same `BIOT` encoder. To form the objective, we want to use the perturbed signal to predict the embedding of the original signal. Thus, an additioanl predictor (i.e., two-layer neural network) is appended for the perturbed signal following (Grill et al., 2020). We use $\mathbf{Z}$ and $\tilde{\mathbf{Z}}$ to denote the real embedding of $\mathbf{S}$ and predicted embedding from $\tilde{\mathbf{S}}$.

$$\mathbf{Z} = \text{BIOT}(\mathbf{S}), \ \ \tilde{\mathbf{Z}} = \text{predictor}(\text{BIOT}(\tilde{\mathbf{S}})). \tag{3}$$

- Finally, contrastive loss (He et al., 2020; Chen et al., 2020) is used on $\mathbf{Z}$ and $\tilde{\mathbf{Z}}$ to form the objective.

$$\mathcal{L} = \text{CrossEntropyLoss}\left(\text{softmax}\left(\langle \mathbf{Z}, \tilde{\mathbf{Z}}^\top \rangle / T\right), \mathbf{I}\right). \tag{4}$$

Here, $T$ represents the temperature ($T = 0.2$ throughout the paper) and $\mathbf{I}$ is an identity matrix. In the implementation, we also apply sample-wise L2-normalization on $\mathbf{Z}$ and $\tilde{\mathbf{Z}}$ before matrix product.

**(4) Supervised Pre-training** aims to pre-train a model by supervised learning on one task and then generalize and fine-tune the encoder on a new task. The goal here is to transfer knowledge among different datasets and gain improvements on the new task compared to training from scratch. Our `BIOT` model allows the new datasets to have mismatched channels and different lengths.

## 3 Experiments

This section shows the strong performance of `BIOT` on several EEG, ECG, and human sensory datasets. Section 3.2, 3.3 compare `BIOT` with baselines on **supervised learning** and **learning with missing** settings. Section 3.4, 3.5, 3.6 show the that `BIOT` can be flexibly **pre-trained on various datasets** (supervised or unsupervised) to improve a new task with different sample formats.

### 3.1 Experimental Setups

**Biosignal Datasets.** We consider the following datasets in the evaluation: (i) **SHHS** (Zhang et al., 2018; Quan et al., 1997) is a large sleep EEG corpus from patients aged 40 years and older. (ii) **PREST** is a large unlabeled proprietary resting EEG dataset; (iii) **Cardiology** (Alday et al., 2020) is a collection of five ECG datasets (initially contains six, and we exclude the PTB-XL introduced below). (iv) The **CHB-MIT** database (Shoeb, 2009) is collected from pediatric patients for epilepsy seizure detection. (v) **IIIC Seizure** dataset is from Ge et al. (2021); Jing et al. (2023) for detecting one of the six ictal-interictal-injury-continuum (IIIC) seizure patterns (OTH, ESZ, LPD, GPD, LRDA, GRDA); (vi) TUH Abnormal EEG Corpus (**TUAB**) (Lopez et al., 2015) is an EEG dataset that has been annotated as normal or abnormal; (vii) TUH EEG Events (**TUEV**) (Harati et al., 2015) is a corpus of EEG that contains annotations of segments as one of six sleep or resting event types: spike and sharp wave (SPSW), generalized periodic epileptiform discharges (GPED), periodic lateralized epileptiform discharges (PLED), eye movement (EYEM), artifact (ARTF) and background (BCKG); (viii) **PTB-XL** (Wagner et al., 2020) is an ECG dataset with 12-lead recordings for diagnosis prediction, and we used it for arrhythmias phenotyping in this paper; (ix) **HAR** (Anguita et al., 2013) is a human action recognition dataset using smartphone accelerometer and gyroscope data.

Table 1: Dataset Statistics

| Datasets | Type (subtype) | # Recordings | Rate | Channels | Duration | # Sample | Tasks |
|---|---|---|---|---|---|---|---|
| SHHS | EEG (sleep) | 5,445 | 125Hz | C3-A2, C4-A1 | 30 seconds | 5,093,522 | Unsupervised pre-training |
| PREST | EEG (resting) | 6,478 | 200Hz | 16 montages | 10 seconds | 5,110,992 | Unsupervised pre-training |
| Cardiology | ECG | 21,264 | 500Hz | 6 or 12 ECG leads | 10 seconds | 495,970 | Unsupervised pre-training |
| CHB-MIT | EEG (resting) | 686 | 256Hz | 16 montages | 10 seconds | 326,993 | Binary (seizure or not) |
| IIIC Seizure | EEG (resting) | 2,702 | 200Hz | 16 montages | 10 seconds | 165,309 | Multi-class (6 seizure types) |
| TUAB | EEG (unknown) | 2,339 | 256Hz | 16 montages | 10 seconds | 409,455 | Binary (abnormal or not) |
| TUEV | EEG (sleep and resting) | 11,914 | 256Hz | 16 montages | 5 seconds | 112,491 | Multi-class (6 event types) |
| PTB-XL | ECG | 21,911 | 500Hz | 12 ECG leads | 5 seconds | 65,511 | Binary (arrhythmias or not) |
| HAR | Wearable sensors | 10,299 | 50Hz | 9 coordinates | 2.56 seconds | 10,299 | Multi-class (6 actions) |

**Dataset Processing.** The first three datasets are used entirely for unsupervised pre-training. The next four datasets are used for supervised learning, and we used the common 16 bipolar montage channels in the international 10-20 system. For CHB-MIT (containing 23 patients), we first use patient 1 to 19 for training, 20,21 for validation, and 22,23 for test. Then, we flip the validation and test sets and conduct the experiments again, and we report the average performance on these two settings. For IIIC seizure, we divide patient groups into training/validation/test sets by 60%:20%:20%. For TUAB and TUEV, the training and test separation is provided by the dataset. We further divide the training patients into training and validation groups by 80%:20%. For PTB-XL, we divide patient groups into training/validation/test sets by 80%:10%:10%. The train and test set of HAR is provided, and we further divide the test patients into validation/test by 50%:50%. For all the datasets, after assigning the patients to either training, validation, or test groups, we will further split the patient's recording to samples, and the sample duration accords to the annotation files. The dataset statistics can be found in Table 1, and we provides more descriptions and processing details in Appendix B.1.

**Baseline.** We consider the following representative models: (i) **SPaRCNet** (Jing et al., 2023) is a 1D-CNN based model with dense residual connections, more advanced than the popular ConvNet (Schirrmeister et al., 2017), CSCM (Sakhavi et al., 2018); (ii) **ContraWR**'s (Yang et al., 2021) encoder model first transforms the biosignals into multi-channel spectrogram and then uses 2D-CNN based ResNet (He et al., 2016); (iii) **CNN-Transformer** (Peh et al., 2022) is superior to CNN-LSTM models (Zhang et al., 2019); (iv) **FFCL** (Li et al., 2022) combines embeddings from CNN and LSTM

encoders for feature fusion; (v) **ST-Transformer** Song et al. (2021) proposes an multi-level EEG transformer for learning spatial (S) and temporal (T) features simultaneously, empirically better than EEGNet Lawhern et al. (2018). Our `BIOT` model trained from scratch is denoted by (vanilla).

**Environments and Settings.** The experiments are implemented by Python 3.9.12, Torch 1.13.1+cu117, Pytorch-lightning 1.6.4 on a Linux server with 512 GB memory, 128-core CPUs and eight RTX A6000 GPUs. All the models are optimized on training set and evaluated on the test set. The best model and hyperparameter combinations are selected based on the validation set. For Table 2 and Table 3, we obtain five sets of results with different random seeds and report the mean and standard deviation values. For Figure 3 and Figure 4, we report the results under three random seeds. More experimental and implementation details can refer to Appendix B.2.

## 3.2 Setting (1) - standard supervised learning

This section shows that `BIOT` is comparable or better than baselines in the supervised learning settings.

- **Four EEG Tasks.** Both CHB-MIT and TUAB are designed to predict binary output, and we use binary cross entropy (BCE) for TUAB and the focal loss (Lin et al., 2017) for CHB-MIT due to its imbalances (around 0.6% positive ratio in training set). We use balanced accuracy (Balanced Acc.), area under precision-recall curve (AUC-PR) and AUROC as the metrics. Both IIIC Seizure and TUEV are multi-class classification tasks with cross entropy loss. We employ Balanced Acc., Cohen's Kappa, and Weighted F1 as the multi-class evaluation. To save space, we only show the performance on CHB-MIT and IIIC Seizure in Table 2 and move the other two to Appendix C.1.

- **ECG and Sensory Tasks.** PTB-XL is formulated as a binary classification on detecting arrhythmias phenotypes. We use the BCE loss and binary evaluation metrics. HAR (classifying actions) uses the cross entropy loss and is evaluated by multi-class metrics. Results are reported in Table 3. We provide the running time comparison of all experiments in Appendix C.4.

Table 2 and 3 show that our model has superior performance over baselines in most tasks, especially on CHB-MIT, IIIC Seizure, and HAR. The reason might be that the frequency features are more useful in these three tasks as our `BIOT` extracts the main features from spectral perspective. SPaRCNet is a strong model among all the baselines except on the CHB-MIT task. The model might be vulnerable in the imbalanced classification setting even with the focal loss. The pre-training models at the end of the tables will be introduced and explained in Section 3.4, 3.6.

Table 2: EEG classification tasks (Results of TUAB and TUEV are in Appendix C.1)

| Models | CHB-MIT (seizure detection) | | | IIIC Seizure (seizure type classification) | | |
|---|---|---|---|---|---|---|
| | Balanced Acc. | AUC-PR | AUROC | Balanced Acc. | Cohen's Kappa | Weighted F1 |
| SPaRCNet (Jing et al., 2023) | $0.5876 \pm 0.0191$ | $0.1247 \pm 0.0119$ | $0.8143 \pm 0.0148$ | $0.5546 \pm 0.0161$ | $0.4679 \pm 0.0228$ | $0.5569 \pm 0.0184$ |
| ContraWR (Yang et al., 2021) | $0.6344 \pm 0.0002$ | $0.2264 \pm 0.0174$ | $0.8097 \pm 0.0114$ | $0.5519 \pm 0.0058$ | $0.4623 \pm 0.0148$ | $0.5486 \pm 0.0137$ |
| CNN-Transformer (Peh et al., 2022) | $0.6389 \pm 0.0067$ | $0.2479 \pm 0.0227$ | $\mathbf{0.8662 \pm 0.0082}$ | $0.5476 \pm 0.0103$ | $0.4481 \pm 0.0139$ | $0.5346 \pm 0.0127$ |
| FFCL (Li et al., 2022) | $0.6262 \pm 0.0104$ | $0.2049 \pm 0.0346$ | $0.8271 \pm 0.0051$ | $0.5617 \pm 0.0117$ | $0.4704 \pm 0.0130$ | $0.5617 \pm 0.0171$ |
| ST-Transformer (Song et al., 2021) | $0.5915 \pm 0.0195$ | $0.1422 \pm 0.0094$ | $0.8237 \pm 0.0491$ | $0.5423 \pm 0.0056$ | $0.4492 \pm 0.0056$ | $0.5440 \pm 0.0014$ |
| (Vanilla) `BIOT` | $\mathbf{0.6640 \pm 0.0037}$ | $\mathbf{0.2573 \pm 0.0088}$ | $0.8646 \pm 0.0030$ | $\mathbf{0.5762 \pm 0.0034}$ | $\mathbf{0.4932 \pm 0.0046}$ | $\mathbf{0.5773 \pm 0.0031}$ |
| Pretrained `BIOT` (PREST) | $0.6942 \pm 0.0431$ | $0.3072 \pm 0.1187$ | $0.8679 \pm 0.0106$ | $0.5787 \pm 0.0066$ | $0.4980 \pm 0.0054$ | $0.5828 \pm 0.0049$ |
| Pretrained `BIOT` (PREST+SHHS) | $0.6788 \pm 0.0036$ | $0.3090 \pm 0.0003$ | $0.8752 \pm 0.0022$ | $\boxed{0.5800 \pm 0.0004}$ | $\boxed{0.5040 \pm 0.0041}$ | $\boxed{0.5878 \pm 0.0015}$ |
| Pretrained `BIOT` (6 EEG datasets) | $\boxed{0.7068 \pm 0.0457}$ | $\boxed{0.3277 \pm 0.0460}$ | $\boxed{0.8761 \pm 0.0284}$ | $0.5779 \pm 0.0087$ | $0.4949 \pm 0.0103$ | $0.5737 \pm 0.0088$ |

1. All models use the same training set of the task, while the pre-trained `BIOT` models are initially pre-trained on other data sources (see Section 3.4, 3.6).
2. **Bold** for the best model (trained from scratch) and box for the best pre-trained models. Running time comparison is in Appendix C.4.

Table 3: ECG and human activity sensory classification tasks

| Models | PTB-XL (arrhythmias phenotype prediction) | | | HAR (humann action recognition) | | |
|---|---|---|---|---|---|---|
| | Balanced Acc. | AUC-PR | AUROC | Balanced Acc. | Cohen's Kappa | Weighted F1 |
| SPaRCNet (Jing et al., 2023) | $0.8275 \pm 0.0047$ | $\mathbf{0.9040 \pm 0.0067}$ | $\mathbf{0.7550 \pm 0.0073}$ | $0.9371 \pm 0.0160$ | $0.9236 \pm 0.0189$ | $0.9365 \pm 0.0155$ |
| ContraWR (Yang et al., 2021) | $0.7532 \pm 0.0561$ | $0.7549 \pm 0.0164$ | $0.5258 \pm 0.1190$ | $0.9068 \pm 0.0164$ | $0.8879 \pm 0.0201$ | $0.9055 \pm 0.0182$ |
| CNN-Transformer (Peh et al., 2022) | $0.6650 \pm 0.0459$ | $0.7175 \pm 0.0558$ | $0.4996 \pm 0.0936$ | $0.8690 \pm 0.0839$ | $0.8273 \pm 0.0953$ | $0.8352 \pm 0.1166$ |
| FFCL (Li et al., 2022) | $0.7034 \pm 0.0052$ | $0.7088 \pm 0.0053$ | $0.5127 \pm 0.0051$ | $0.8519 \pm 0.0148$ | $0.8216 \pm 0.0177$ | $0.8508 \pm 0.0138$ |
| ST-Transformer (Song et al., 2021) | $0.7238 \pm 0.0083$ | $0.7775 \pm 0.0153$ | $0.6003 \pm 0.0179$ | $0.9336 \pm 0.0063$ | $0.9213 \pm 0.0076$ | $0.9337 \pm 0.0068$ |
| (Vanilla) `BIOT` | $\mathbf{0.8315 \pm 0.0008}$ | $0.8978 \pm 0.0020$ | $0.7493 \pm 0.0167$ | $\mathbf{0.9461 \pm 0.0134}$ | $\mathbf{0.9351 \pm 0.0160}$ | $\mathbf{0.9458 \pm 0.0136}$ |
| Pretrained `BIOT` (Cardiology-6) | $0.8350 \pm 0.0073$ | $0.9128 \pm 0.0094$ | $\boxed{0.7671 \pm 0.0116}$ | / | / | / |
| Pretrained `BIOT` (Cardiology-12) | $\boxed{0.8421 \pm 0.0030}$ | $\boxed{0.9221 \pm 0.0075}$ | $0.7659 \pm 0.0076$ | / | / | / |

\* **Bold** for the best model. All models use the same training set of the task. The Pretrained `BIOT` (Cardiology-6) and Pretrained `BIOT` (Cardiology-12) are pre-trained on Cardiology data (see Section 3.4), and they do not apply to HAR data (due to different biosignal types).

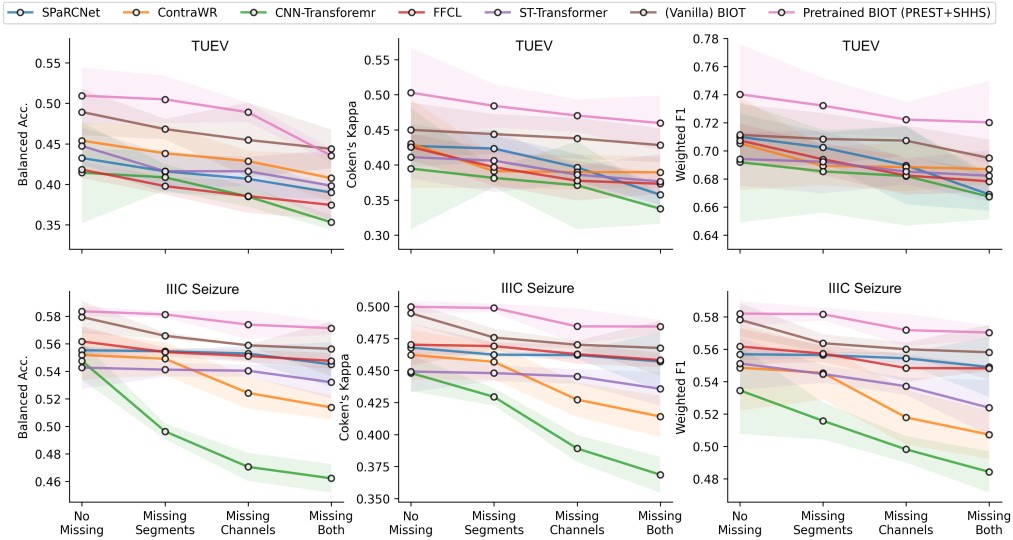

Figure 3: Supervised learning with missing channels or segments (on TUEV and IIIC Seizure)

### 3.3 Setting (2) - learning with missing channels and segments

The section simulates the TUEV dataset (16 channels and 5s per sample) and IIIC Seizure (16 channels and 10s per sample) to mimic the setting of *supervised learning with missing channels and segments*, which shows the strong performance of `BIOT`. We consider three missing cases:

- **Missing segments**: Randomly mask out $a$ segments (each segment spans for 0.5 seconds), $a = 0, 1, 2, 3, 4, 5$ with equal probability. The segment masking is applied separately for each channel.
- **Missing channels**: Randomly mask out $b$ channels, $b = 0, 1, 2, 3, 4$ with equal probability. We assume that the masking will not alter the underlying labels (the same assumption for other cases).
- **Missing both channels and segments**: Combine both masking strategies simultaneously.

To enable the baseline models compatible with the setting, we use all zeros to impute the masked regions. The comparison is plotted in Figure 3, which shows that (i) all models decrease the performance with more missings while `BIOT` and the pre-trained `BIOT` are less impacted (especially on Kappa and Weighted F1); (ii) "Missing channels" affects the performance more than "Missing segments", which makes sense as segment masking still preserves information from all channels.

### 3.4 Setting (3) - unsupervised pre-training

This section shows that `BIOT` enables unsupervised pre-training on existing datasets with various formats. Note that, all the pre-trained models in this paper have a similar scale ($\sim$3.3 million parameters). We defer the study of "scaling effect" of pre-trained `BIOT` to future work.

- **Pre-trained (PREST)**: This model is pre-trained on 5 million resting EEG samples (PREST) with 2,048 as the batch size. We save the pre-trained model at the 100-th epoch.
- **Pre-trained (PREST+SHHS)**: This model is jointly pre-trained on 5M PREST and 5M SHHS EEG samples. Though two datasets have different sample formats, our model is able to use them together. Also, we use 2048 as the batch size and save model at the 100-th epoch.
- **Pre-trained (Cardiology-12)** is jointly pre-trained on raw data of five datasets in Cardiology corpus (details in Appendix B.1). We use 1024 as batch size and save model at the 100-th epoch.
- **Pre-trained (Cardiology-6)** is pre-trained similarly as Pre-trained (Cardiology-12), while we only utilize the first 6 ECG leads. By contast, Pre-trained (Cardiology-12) uses full 12 leads.

We fine-tune the first two pre-trained EEG models on four EEG tasks and append the results to Table 2 (also in Appendix C.1). We fine-tune the last two pre-trained ECG models on PTB-XL datasets in Table 3. The pre-trained models can be seamlessly applied to various downstream tasks with different sample formats. Results show that the pre-trained models can improve the performance consistently after fine-tuning on the training set of downstream tasks in Table 2 and Table 3.

### 3.5 Setting (4) - supervised pre-training on other tasks

This section shows that `BIOT` allows knowledge transfer from one task to another similar task with different sample formats. We pre-train on the training set of CHB-MIT, IIIC Seizure, TUAB and fine-tunes on TUEV (which has 16 channels and 5s duration). All datasets use 200Hz sampling rate. We design three sets of configurations for the pre-trained datasets: **Format (i)** uses the first 8 channels and 10s duration; **Format (ii)** uses the full 16 channels but only the first 5s recording; **Format (iii)** uses full 16 channels and full 10s recording. During fine-tuning, we then remove the prediction layers from these pre-trained model and add a new prediction layer to fit the TUEV dataset.

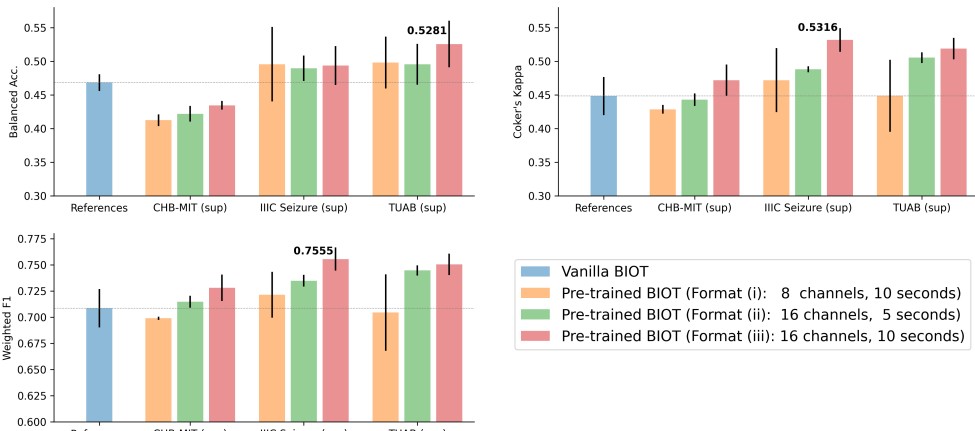

Figure 4: Fine-tuned on TUEV from different supervised pre-trained models (best number in **bold**). Similar supervised fine-tuning analysis on CHB-MIT dataset is shown in Appendix C.2.

The fine-tuning results on TUEV are shown in Figure 4 where we also add the vanilla `BIOT` for reference. We find that (i) the model pre-trained on IIIC Seizure and TUAB are generally beneficial for the event classification task on TUEV. The reason might be that TUAB and TUEV are both recorded from Temple University and share some common information, while IIIC seizure and TUEV are both related to seizure detection and may share some latent patterns. (ii) More pre-training data leads to better results in the downstream task: though the pre-training configuration (16 channels, 5 seconds) aligns better with the TUEV data formats, the results show that configuration of (16 channels, 10 seconds) encodes longer duration and works consistently better. (iii) Compared to the TUEV results in Appendix C.1, we also find that oftentimes the supervised pre-training (e.g., on IIIC seizure or TUAB) can be more effective than unsupervised pre-training (e.g., on SHHS and PREST).

### 3.6 Pre-trained on all EEG datasets

In this section, we show that `BIOT` can leverage all six EEG resources considered in the paper. We obtain a **Pre-trained (6 EEG datasets)** model by loading the Pre-trained (PREST+SHHS) model and further train it on the training sets of CHB-MIT, IIIC Seizure, TUAB, and TUEV. We add separate classification layers for four tasks. Essentially, this model is pre-trained on all six EEG datasets. To use the model, we still fine-tune it on the training set of downstream tasks and append the results to Table 2 and Appendix C.1. Apparently, Pre-trained (6 EEG datasets) outperforms the vanilla `BIOT` and is better than the unsupervised and the supervised pre-trained `BIOT` models in most cases.

## 4 Conclusions and Discussions

This paper proposes a new biosignal transformer model (`BIOT`) that learns embeddings for biosignals with various formats. `BIOT` can enable effective knowledge transfer across different data and allow joint training on multiple sources. We conduct extensive evaluations on nine biosignal datasets and show that our `BIOT` is flexible and effective in various cross-data learning settings. Future efforts can explore the different types of biosignals and pre-training an all-in-one unified biosignal model. We hope our work can inspire more follow-up researches of large foundational models for biosignals.

**Acknowledgments**

This work was supported by NSF award SCH-2205289, SCH-2014438, and IIS-2034479. This project has been funded by the Jump ARCHES endowment through the Health Care Engineering Systems Center.

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

# A  Notaion Table

The listed the notations used in the main text in the following Table 4.

Table 4: Notations used in BIOT

| Symbols | Descriptions |
|---|---|
| $I \in \mathbb{N}^+$ | the number of channels in biosignal, such as 16 |
| $J \in \mathbb{R}^+$ | the length of biosignals, such as $10s \times 200Hz$ (use a complete sample for ease of notation) |
| $\mathbf{S} \in \mathbb{R}^{I \times J}$ | the mulit-channel biosignal |
| $\mathbf{S}[i] \in \mathbb{R}^J$ | the $i$-th channel of the biosignal. |
| $r \in \mathbb{R}^+$ | the sampling rate, such as 200Hz |
| $t \in \mathbb{R}^+$ | the length of the biosignal token, such as $1s \times 200Hz$ |
| $p \in \mathbb{R}^+$ | the overlap length between two neighboring tokens, such as $0.5s \times 200Hz$, $t > p$ |
| $k \in \mathbb{N}^+$ | the $k$-th token in the $i$-th channel is denoted by $\mathbf{S}[i, (t-p)(k-1) : (t-p)(k-1) + t]$ which has length $t$ |
| $K \in \mathbb{N}^+$ | max number of tokens in one channel |
| $N \in \mathbb{N}^+$ | the number of tokens in tokenized biosignal "sentence" |
| $l_1 \in \mathbb{N}^+$ | the dimension of the token embedding |
| $\mathbf{X} \in \mathbb{R}^{N \times l_1}$ | the tokenized biosignal "sentence" |
| $l_2 \in \mathbb{N}^+$ | the new dimensions of tokens after self-attention module |
| $\mathbf{W}^K, \mathbf{W}^V, \mathbf{W}^Q \in \mathbb{R}^{l_1 \times l_2}$ | the key, value, and query matrices in self-attention module |
| $\mathbf{E}, \mathbf{F} \in \mathbb{R}^{d \times N}$ | the low-rank projection matrices |
| $d \in \mathbb{N}^+$ | the reduced rank for self-attention matrices, $d \ll N$ |
| $\mathbf{H} \in \mathbb{R}^{N \times l_2}$ | the output of the self-attention module |
| $\tilde{\mathbf{S}} \in \mathbb{R}^{I \times J}$ | the perturbed biosignal sample |
| $\mathbf{Z}, \tilde{\mathbf{Z}}$ | the real and predicted embeddings of $\mathbf{S}$ |
| $T$ | the temperature hyperparameter in the contrastive loss |
| $\mathbf{I}$ | the identity matrix in the contrastive loss, has the size of (batch size, batch size) |
| $a \in [0, 1, 2, 3, 4, 5]$ | the discrete uniformly distributed variable for the number of masked segments in Section 3.3 |
| $b \in [0, 1, 2, 3, 4]$ | the discrete uniformly distributed variable for the number of masked channels in Section 3.3 |

# B  Details of Datasets and Experimental Settings

## B.1  More for Datasets and Processings

We provide more descriptions on each dataset in this section.

**For EEG datasets.** First, the 16 derivations (in 10-20 international system) are "FP1-F7", "F7-T7", "T7-P7", "P7-O1", "FP2-F8", "F8-T8", "T8-P8", "P8-O2", "FP1-F3", "F3-C3", "C3-P3", "P3-O1", "FP2-F4", "F4-C4", "C4-P4", "P4-O2".

- Sleep Heart Health Study (**SHHS**) (Zhang et al., 2018; Quan et al., 1997) is a multi-center cohort study from the National Heart Lung & Blood Institute assembled to study sleep-disordered breathing, which contains 5,445 recordings. The data is accessible upon request in their website [2]. Each recording has 14 Polysomnography (PSG) channels, and the recording frequency is 125.0 Hz. We use the C3/A2 and C4/A1 EEG channels. The dataset is released with sleep annotations. We use the existing codes [3] and split each recordings into 30-second samples. In this study, we use SHHS samples for unsupervised pre-training without the original labels.

- **PREST** is a private dataset recorded in hospital sleep lab, primarily for seizure and abnormal EEG detection purpose (such as spikes). The local IRB waived the requirement for informed consent for this retrospective analysis of EEG data. We follow the clinician's instructions and split each recordings into 10 seconds without labels. In the experiment, we use it for EEG model pre-training.

- The **CHB-MIT** database [4] (Shoeb, 2009) is publicly available, which is collected at the Children's Hospital Boston, consists of EEG recordings from pediatric subjects with intractable seizures. The dataset is under Open Data Commons Attribution License v1.0 [5] and is used to predict whether the EEG recordings contain seizure signals. Each recording initially contains 23 bipolar channels and

---

[2]https://sleepdata.org/datasets/shhs

[3]https://github.com/ycq091044/ContraWR/tree/main/preprocess

[4]https://physionet.org/content/chbmit/1.0.0/

[5]https://physionet.org/content/chbmit/view-license/1.0.0/

we select the 16 standard derivations in the experiments. We utilize the existing preprocessing [6] and follow the typical practices to further split each recordings into 10-second non-overlapping samples by default. Since the dataset is highly imbalanced, we use 5 seconds as overlaps to split the seizure regions (which could potentially double the positive samples). After processing, the positive ratio in the training set is around 0.6%.

- **IIIC Seizure** is requested from Ge et al. (2021); Jing et al. (2023), and we follow the license and usage statements in Jing et al. (2023). The samples follow 16 derivations and span 10-second signals at 200Hz. This dataset is used for predicting one of the six classes: lateralized periodic discharges (LPD), generalized periodic discharges (GPD), lateralized rhythmic delta activity (LRDA), generalized rhythmic delta activity (GRDA), Seizure types, and Other.

- TUH Abnormal EEG Corpus (**TUAB**) (Lopez et al., 2015) and TUH EEG Events (**TUEV**) (Harati et al., 2015) is accessible upon request at Temple University Electroencephalography (EEG) Resources [7]. We process both datasets to follow the 16 EEG derivations.

**For ECG datasets.** We use the Cardiology collection to pre-train the ECG models and apply it on downstream supervisd PTB-XL task.

- The **Cardiology** collection (Alday et al., 2020) is publicly available at physionet [8], which was used in the PhysioNet/Computing in Cardiology Challenge 2020. This collection is under Creative Commons Attribution 4.0 International Public License [9]. In this study, we use five sets from the training portion of the collection (It has in total six sets. Another one overlaps with the PTB-XL dataset, and thus we drop it for pre-training), which contains recordings from CPSC2018 (6,877 recordings), CPSC2018Extra (China 12-Lead ECG Challenge Database – unused CPSC 2018 data, 3,453 recordings), St Petersburg Incart (12-lead Arrhythmia Database, 74 recordings), PTB (Diagnostic ECG Database, 516 recordings), Georgia (12-Lead ECG Challenge Database, 10,344 recordings). For preprocessing, we extract 10-second samples from each recording with 0.5s as the overlapping window (for obtaining more unsupervised trianing corpus). All the samples are merged together as an unsupervised pre-training ECG corpus of nearly 0.5 million samples. We pre-train a Pre-trained `BIOT` (Cardiology-12) on all the channels and a Pre-trained `BIOT` (Cardiology-6) on the first 6-channels of all samples. The sample sizes are different from the below PTB-XL dataset.

- Physikalisch-Technische Bundesanstalt (**PTB-XL**) [10] (Wagner et al., 2020) is a publicly available large dataset of 12-lead ECGs from 18885 patients. It is under the Creative Commons Attribution 4.0 International Public License [11]. The raw waveform data was annotated by up to two cardiologists, who assigned potentially multiple ECG statements to each record up to 27 diagnoses: 1:1st degree AV block, 2:Atrial fibrillation, 3:Atrial flutter, 4:Bradycardia, 5:Complete right bundle branch block, 6:Incomplete right bundle branch block, 7:Left anterior fascicular block, 8:Left axis deviation, 9:Left bundle branch block, 10:Low QRS voltages, 11:Nonspecific intraventricular conduction disorder, 12:Pacing rhythm, 13:Premature atrial contraction, 14:Premature ventricular contractions, 15:Prolonged PR interval, 16:Prolonged QT interval, 17:Q wave abnormal, 18:Right axis deviation, 19:Right bundle branch block, 20:Sinus arrhythmia, 21:Sinus bradycardia, 22:Sinus rhythm, 23:Sinus tachycardia, 24:Supraventricular premature beats, 25:T wave abnormal, 26:T wave inversion, 27:Ventricular premature beats. We following clinical knowledges and further groups them into six broader categories: Arrhythmias, Bundle branch blocks and fascicular blocks, Axis deviations, Conduction delays, Wave abnormalities, Miscellaneous. Each recordings can be associated to multiple categories. In this paper, we conduct the "Arrhythmias" phenotyping prediction task. If the recordings have at least one diagnosis belonging to the Arrhythmias group, then we label them as positive, otherwise as negative.

**For human activity sensory data.** Human activity recognition (**HAR**) dataset [12] (Anguita et al., 2013) is publicly available at UCI machine learning repository. The data is collected from smartphone accelerometer and gyroscope data with 3D coordinates to detect six actions: walking, walking

---

[6]https://github.com/bernia/chb-mit-scalp

[7]https://isip.piconepress.com/projects/tuh_eeg/html/downloads.shtml

[8]https://physionet.org/content/challenge-2020/1.0.2/

[9]https://physionet.org/content/challenge-2020/view-license/1.0.2/

[10]https://physionet.org/content/ptb-xl/1.0.1/

[11]https://physionet.org/content/ptb-xl/view-license/1.0.1/

[12]https://archive.ics.uci.edu/ml/datasets/human+activity+recognition+using+smartphones

upstairs, walking downstairs, sitting, standing, laying. The samples are already splitted and provided in the original datasets.

## B.2 More for Experimental Settings

For model implementation, the SPaRCNet code is requested from the authors (Jing et al., 2023), the ContraWR code is downloaded and modified upon the github [13], CNN-Transformer is easily implemented following the Fig. 3 of the original paper (Peh et al., 2022), FFCL (Li et al., 2022) combines a CNN model and a LSTM model for learning separete representations and then merges them before the final prediction layer, the implementation of ST-Transformer refer to this repo [14]. The linear-complexity attention module is referred to this repo [15] in our BIOT implementation.

For all EEG tasks, we resample the datasets into 200Hz. The ECG tasks use 500Hz, and the HAR tasks use 50Hz by default. For each specific tasks, we have to adjust the baseline model architectures (e.g, number of layers, input channel sizes, etc) accordingly since the input data have various formats. While for our BIOT, we only adjust the fft size based on their sampling rate (200Hz for EEG, 1000Hz for ECG, 100Hz for HAR) and use 0.5s, 0.2s, and 0.1s as the hop length (i.e., overlaps) in three signal types. These model configurations are chosen by testing several combinations based on the validation performance and we select the best one. For our BIOT model, we use 8 as the number of head, 4 as the number of transformer layers, and $T = 2$ as the temperature in unsupervised pre-training by default. We use the Adam optimizer with learning rate $1 \times 10^{-3}$ and $1 \times 10^{-5}$ as the coefficient for L2 regularization by default. We use the pytorch lightning framework (with 100 as the max epoch) to handle the training, validation, and test pipeline by setting AUROC as the monitoring metirc for binary classification and Coken's Kappa as the monitoring metric for multi-class classification in the validation. More details can refer to our Supplementary codes. Below, we provide the definition of each metric used in the paper.

**Balanced Accuracy** is defined as the average of recall obtained on each class. It is used for both binary classification and multi-class classification.

**AUC-PR** is the area under the precision recall (PR) curve for binary classification task.

**AUROC** is the area under the ROC curve, summarizing the ROC curve into an single number that describes the performance of a model for multiple thresholds at the same time. It is used for binary classification.

**Coken's Kappa** is a statistic that measures inter-annotator agreement, which is usually used for imbalanced multi-class classification task. The calculation can refer to sklearn metrics [16].

**Weighted F1** is used for multi-class classification in this paper, which is a weighted average of individual F1-scores from each class, with each score weighted by the number of samples in the corresponding class.

# C Additional Results

This section provides additional experimental results to support claims in the main paper.

## C.1 Additional Experiments on TUEV and TUAB

We have provided the supervised learning results on EEG dataset IIIC Seizure and CHB-MIT in the main text. For completeness, we provide similar comparison results on TUAB and TUEV below in Table 5 6, which show a similar trend that our BIOT shows better performance against baseline models, and the pre-trained BIOT models can bring significant improvements on two downstream tasks, especially on TUEV. For TUEV, we also append the results of all different pre-trained models (e.g., train from scratch, supervised training, unsupervised training, etc) in the end in Table 6.

---

[13]https://github.com/ycq091044/ContraWR
[14]https://github.com/eeyhsong/EEG-Transformer
[15]https://github.com/lucidrains/linear-attention-transformer
[16]https://scikit-learn.org/stable/modules/generated/sklearn.metrics.cohen_kappa_score.html

Table 5: Additional Supervised Learning Results on TUAB

| Models | TUAB (abnormal detection) | | |
|---|---|---|---|
| | Balanced Acc. | AUC-PR | AUROC |
| SPaRCNet | $0.7896 \pm 0.0018$ | $0.8414 \pm 0.0018$ | $0.8676 \pm 0.0012$ |
| ContraWR | $0.7746 \pm 0.0041$ | $0.8421 \pm 0.0104$ | $0.8456 \pm 0.0074$ |
| CNN-Transformer | $0.7777 \pm 0.0022$ | $0.8433 \pm 0.0039$ | $0.8461 \pm 0.0013$ |
| FFCL | $0.7848 \pm 0.0038$ | $0.8448 \pm 0.0065$ | $0.8569 \pm 0.0051$ |
| ST-Transformer | $0.7966 \pm 0.0023$ | $0.8521 \pm 0.0026$ | $\mathbf{0.8707 \pm 0.0019}$ |
| (Vanilla) BIOT | $\mathbf{0.7925 \pm 0.0035}$ | $\mathbf{0.8707 \pm 0.0087}$ | $0.8691 \pm 0.0033$ |
| Pre-trained BIOT (PREST) | $0.7907 \pm 0.0050$ | $0.8752 \pm 0.0051$ | $0.8730 \pm 0.0021$ |
| Pre-trained BIOT (PREST+SHHS) | $\boxed{0.8019 \pm 0.0021}$ | $0.8749 \pm 0.0054$ | $0.8739 \pm 0.0019$ |
| Pre-trained BIOT (6 EEG datasets) | $0.7959 \pm 0.0057$ | $\boxed{0.8792 \pm 0.0023}$ | $\boxed{0.8815 \pm 0.0043}$ |

\* **Bold** for the best model (trained from scratch) and $\boxed{\text{box}}$ for the best pre-trained models.

Table 6: Additional Supervised Learning Results on TUEV (All-in-one-table comparison)

| Models | TUEV (event type classification) | | |
|---|---|---|---|
| | Balanced Acc. | Coken's Kappa | Weighted F1 |
| **(Training from scratch in Section 3.2** | | | |
| SPaRCNet | $0.4161 \pm 0.0262$ | $0.4233 \pm 0.0181$ | $0.7024 \pm 0.0104$ |
| ContraWR | $0.4384 \pm 0.0349$ | $0.3912 \pm 0.0237$ | $0.6893 \pm 0.0136$ |
| CNN-Transformer | $0.4087 \pm 0.0161$ | $0.3815 \pm 0.0134$ | $0.6854 \pm 0.0293$ |
| FFCL | $0.3979 \pm 0.0104$ | $0.3732 \pm 0.0188$ | $0.6783 \pm 0.0120$ |
| ST-Transformer | $0.3984 \pm 0.0228$ | $0.3765 \pm 0.0306$ | $0.6823 \pm 0.0190$ |
| (Vanilla) BIOT | $0.4682 \pm 0.0125$ | $0.4482 \pm 0.0285$ | $0.7085 \pm 0.0184$ |
| **(Unsuperived pre-trained models in Section 3.4):** | | | |
| Pre-trained BIOT (PREST) | $0.5207 \pm 0.0285$ | $0.4932 \pm 0.0301$ | $0.7381 \pm 0.0169$ |
| Pre-trained BIOT (PREST+SHHS) | $0.5149 \pm 0.0292$ | $0.4841 \pm 0.0309$ | $0.7322 \pm 0.0196$ |
| **(Supervised pre-trained models in Section 3.5):** | | | |
| Pre-trained BIOT (pre-trained on CHB-MIT with 8 channels and 10s) | $0.4123 \pm 0.0087$ | $0.4285 \pm 0.0065$ | $0.6989 \pm 0.0015$ |
| Pre-trained BIOT (pre-trained on CHB-MIT with 16 channels and 5s) | $0.4218 \pm 0.0117$ | $0.4427 \pm 0.0093$ | $0.7147 \pm 0.0058$ |
| Pre-trained BIOT (pre-trained on CHB-MIT with 16 channels and 10s) | $0.4344 \pm 0.0065$ | $0.4719 \pm 0.0231$ | $0.7280 \pm 0.0126$ |
| Pre-trained BIOT (pre-trained on IIIC seizure with 8 channels and 10s) | $0.4956 \pm 0.0552$ | $0.4719 \pm 0.0475$ | $0.7214 \pm 0.0220$ |
| Pre-trained BIOT (pre-trained on IIIC seizure with 16 channels and 5s) | $0.4894 \pm 0.0189$ | $0.4881 \pm 0.0045$ | $0.7348 \pm 0.0056$ |
| Pre-trained BIOT (pre-trained on IIIC seizure with 16 channels and 10s) | $0.4935 \pm 0.0288$ | $0.5316 \pm 0.0176$ | $0.7555 \pm 0.0111$ |
| Pre-trained BIOT (pre-trained on TUAB with 8 channels and 10s) | $0.4980 \pm 0.0384$ | $0.4487 \pm 0.0535$ | $0.7044 \pm 0.0365$ |
| Pre-trained BIOT (pre-trained on TUAB with 16 channels and 5s) | $0.4954 \pm 0.0305$ | $0.5053 \pm 0.0079$ | $0.7447 \pm 0.0049$ |
| Pre-trained BIOT (pre-trained on TUAB with 16 channels and 10s) | $0.5256 \pm 0.0348$ | $0.5187 \pm 0.0160$ | $0.7504 \pm 0.0102$ |
| **(Supervised + unsupervised pre-trained model in Section 3.6):** | | | |
| Pre-trained BIOT (6 EEG datasets) | $0.5281 \pm 0.0225$ | $0.5273 \pm 0.0249$ | $0.7492 \pm 0.0082$ |

## C.2  Additional Experiments on CHB-MIT

This section performs a similar experiment on CHB-MIT, similar to Section 3.2. We pre-train on the training set of IIIC Seizure (which has 16 channels and 10s duration), TUAB (which has 16 channels and 10s duration), TUEV (which has 16 channels and 5s duration) and fine-tunes on CHB-MIT (which has 16 channels and 10s duration). All datasets use 200Hz sampling rate. We design five sets of configurations for the pre-trained datasets: **Format (i)** uses the first 8 channels and 10s duration; **Format (ii)** uses the full 16 channels but only the first 5s recording; **Format (iii)** uses full 16 channels and full 10s recording; **Format (iv)** uses 8 channels and 5s recording, and **Format (v)** uses full 16 channels and 2.5s recording. The last two are only for the TUEV dataset. During fine-tuning, we then remove the prediction layers from these pre-trained model and add a new prediction layer to fit the CHB-MIT dataset.

The results are reported in Figure 5, which shows that the supervised pre-training on both IIIC seizure and TUEV can help improve the downstream performance on CHB-MIT task compared to training from scratch. The reason is that IIIC Seizure is on multiple seizure type classification while CHB-MIT is on binary "seizure or not" classification, and the context of both tasks are fairly related. Although TUEV is not entirely on seizure related classification, some classes in TUEV are seizure subtypes (such as GPED, PLED), and thus its supervisd pre-trained models can also bring benefits for the CHB-MIT task.

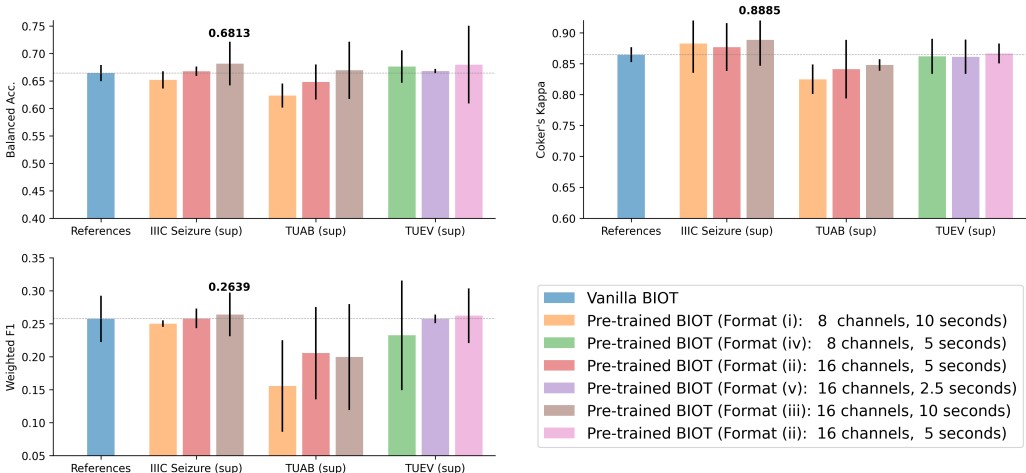

Figure 5: Fine-tuned on CHB-MIT from different supervised pre-trained models. IIIC Seizure and TUAV datasets follow Format (i)(ii)(iii), while TUEV follows the Format (iv)(v)(ii).

## C.3 Ablation Studies on Hyperparameters

This section provides ablation studies on three hyperparameters in data processing: target sampling rate $r$, token length $t$, and the overlap size $p$ between two neighboring tokens. We use two EEG datasets as example: IIIC Seizure and TUAB. The default configuration in the main paper is **(1) sampling**: $r = 200Hz$, **(2) token length**: $t = 1s \times r$, **(3) overlaps**: $p = 0.5s \times r$ as reference.

We also show an illustration for tokenization with overlap in Figure 6.

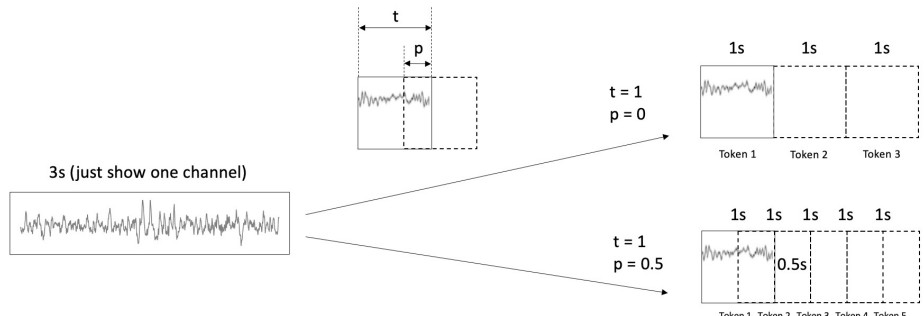

Figure 6: Illustration on tokenization with token length $t$ and overlap $p$. The example shows one channel with 3 seconds. The configuration of $(t = r, p = 0)$ gives 3 tokens while the configuration of $(t = r, p = 0.5r)$ gives 5 tokens. Different configurations lead to different lengths.

### C.3.1 Ablation Study on Target Sampling Rate $r$

In this experiment, we fix the coefficient "1" and "0.5" in (2)(3) and conduct ablation study on the target sampling rate $r$. The original IIIC Seizure data is at 200Hz and the TUAB data is at 256Hz. For IIIC Seizure, we vary the sampling rate to 26Hz, 50Hz, 100Hz, 150Hz, and 200Hz. For TUAB, we vary the sampling rate to 50Hz, 100Hz, 150Hz, 200Hz, 250Hz, and 300Hz. We use the tool $scipy.signal.resample$. The evaluations are conducted under three different random seeds and the mean and standard deviation values are reported.

For IIIC Seizure, we can observe that a higher sampling rate could give slightly better performance, especially on balanced acc. and coken's kappa. The reason is that higher sampling rate can preserve more detailed (high-frequency) biosignal information. The results on TUAB shows that the performances increase and then decrease slightly during increasing the sampling rate. We guess that with the increasing of $r$, initially the performance improves due to obtaining more information. Later,

higher sampling rate does not bring more benefits but unnecessary frequency bands might incur some noise. We also conjecture that different tasks might have diverse sensitivity to the the frequency bands. For example, the task on IIIC seizure is to classify different seizure types, which may need to capture minor clues from high-frequency waves (such as Gamma waves (50-100Hz)), while the TUAB dataset is for abnormal detection, and using brain waves under 50Hz might be enough for the task. In sum, the target sampling rate $r$ should be selected based on the predicting targets.

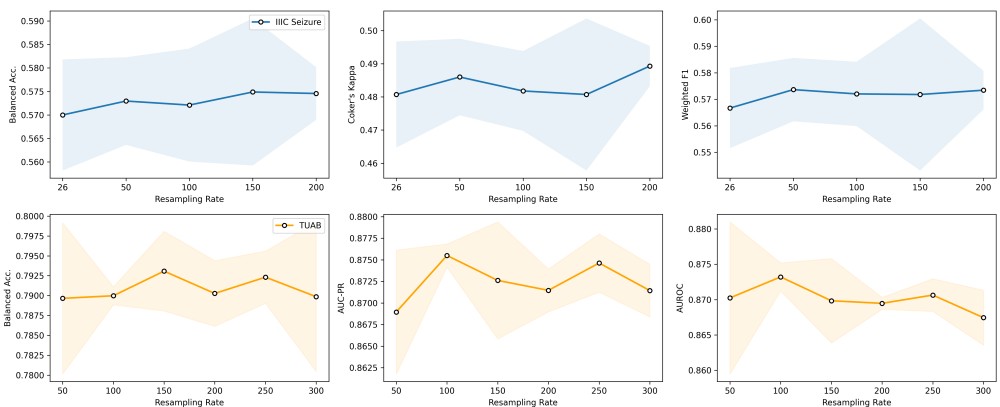

Figure 7: Ablation Study on Target Sampling Rate $r$

### C.3.2 Ablation Study on Token Lengths $t$

In this experiments, we fix (1)(3) and conduct ablation study on the coefficient in token length $t$. Both datasets have 10s as the entire sample length and 0.5s as the overlap lengths. For both of the datasets, we vary the token length coefficient to 0.75s, 1s, 1.5s, 2s, 2.5s, 5s. The evaluations are conducted under three different random seeds and the mean and standard deviation values are reported.

For each configuration, we also set the fft size to match the token length, which means that 5s token duration will bring more frequency information. However, we find that by increasing the token lengths, the model performance starts to decrease. Model performances on IIIC Seizure starts to decrease after 1s while the performance on TUAB decreases after 2s. The reason could be that (i) longer token length (i.e., frequency bands) do not provide extra benefits for learning the tasks; (ii) given the increasing token lengths $t$, the total biosignal "sentence" length, which is $\frac{J-t}{t-p} + 1 = \frac{10-t}{t-0.5} + 1$, will decrease (here, $J = 10r$ is the channel duration, $t$ is the token length, $p = 0.5r$ is the overlapping length). For example, with $t = 5r$ as the token lengths, the number of tokens becomes 2 per channel while the number of tokens per channel is 19 in the default configuration with $t = r$. The performance drops is due to transformer models will be less beneficial in shorter "sentence"s.

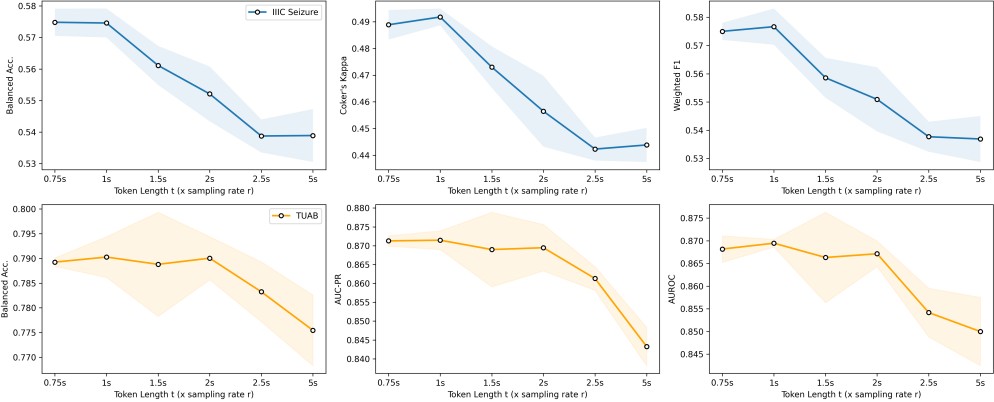

Figure 8: Ablation Study on Token Lengths $t$

### C.3.3 Ablation Study on Overlapping Lengths $p$

In this experiments, we fix (1)(2) and conduct ablation study on the coefficient in the overlap length $p$. Both datasets have 10s as the entire sample length and 1s as the token lengths. For both of the datasets, we vary the overlap length coefficient to 0.875s, 0.75s, 0.5s, 0.25s, 0s. The evaluations are conducted under three different random seeds and the mean and standard deviation values are reported.

Based on the "sentence" length formula $\frac{J-t}{t-p}+1$, smaller overlap length $p$ will decrease the "sentence" length. On both datasets, we find that larger overlap can brings better results due to that the biosignal "sentence" becomes longer, and Transformer models can be more beneficial in the cases. Another reason is that with larger overlaps, neighboring tokens can capture more transitioning information and help the transformer model to better capture the temporal dynamics.

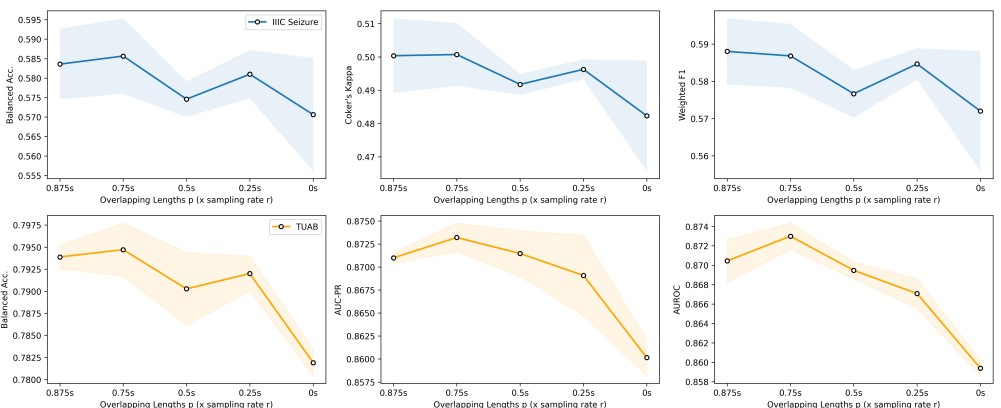

Figure 9: Ablation Study on Overlapping Lengths $p$ Between Tokens

### C.4 Running Time Comparison

This section compares the running time of all models on all supervised learning tasks (CHB-MIT, TUAB, IIIC Seizure, TUEV, PTB-XL, HAR). When recording the running time, we duplicated the environment mentioned in Section 3.2, stopped other programs, and ran all the models one by one on one GPU. We record the first 10 epochs of all models and report the per epoch mean and standard deviation of the time cost in Table 7. In the setting, we use 512 as the batch size for CHB-MIT, TUAB, TUEV, and PTB-XL, 256 as batch size for IIIC Seizure, and 64 as batch size for HAR.

The results show that our model has a similar running time profile as CNN-Transformer and the ST-Transformer models. These two baselines and our `BIOT` are all transformer based models and have the same number of heads and layers. `BIOT` uses linear self-attention module which should be faster than CNN-Transformer and the ST-Transformer, however, our sequence structure are also generally longer due to the flattening transformation. We also find that the CNN models (SPaRCNet and ContraWR) are faster than the transformer based models in our experiments.

In all, the running time of all models are in the same magnitude, and the actual running time on certain applications can vary due to the signal length, number of channels, selected number of CNN or transformer layers or attention heads. The reported running time comparisons here work as references using our selected model architectures. We think the running time of `BIOT` is acceptable given its decent performance.

Table 7: Running time comparison (seconds per epoch)

| Model | CHB-MIT | TUAB | IIIC Seizure | TUEV | HAR | PTB-XL |
|---|---|---|---|---|---|---|
| SPaRCNet | $32.4417 \pm 0.9952$ | $17.4635 \pm 1.0861$ | $24.8728 \pm 2.9339$ | $7.1237 \pm 1.8801$ | $1.9485 \pm 0.0926$ | $7.3850 \pm 0.0599$ |
| ContraWR | $24.7308 \pm 0.7905$ | $13.4650 \pm 0.4554$ | $14.9449 \pm 0.9323$ | $5.3337 \pm 0.2003$ | $2.3235 \pm 0.1308$ | $5.1978 \pm 0.0366$ |
| CNN-Transforemr | $53.3355 \pm 1.8880$ | $25.8983 \pm 0.7969$ | $25.7742 \pm 1.2990$ | $9.0415 \pm 0.2501$ | $3.0207 \pm 0.0463$ | $7.9851 \pm 0.0400$ |
| FFCL | $43.6103 \pm 0.7927$ | $21.6868 \pm 0.5668$ | $23.7682 \pm 1.0757$ | $7.0863 \pm 0.1333$ | $2.6649 \pm 0.0973$ | $6.5135 \pm 0.0228$ |
| ST-Transformer | $50.4725 \pm 0.9978$ | $24.5641 \pm 0.4770$ | $26.3251 \pm 3.2320$ | $7.9027 \pm 0.0758$ | $2.7954 \pm 0.0698$ | $11.1495 \pm 0.0200$ |
| (Vanilla) BIOT | $55.5780 \pm 0.4229$ | $25.2788 \pm 0.1200$ | $25.4500 \pm 4.0835$ | $8.1812 \pm 0.1228$ | $2.9560 \pm 0.0563$ | $12.1791 \pm 0.0452$ |

### C.5 Discussion on handling long recordings and multiple channels

Currently, the BIOT model already have two designs to handle long recordings and more channels: (i) the current model uses linear complexity transformer, so that the complexity scales linearly with sample lengths and channel sizes; (ii) we can remove the token overlaps and enlarge the token sizes to reduce the token numbers (i.e., "sentence" length). With minor adjustments, the BIOT model can better handle long recordings and multiple channels. For long recordings, we could segment the recordings into 10-30s samples and then apply our BIOT on each sample and finally use a top level LSTM or Transformer to learn sequence embedding. For multiple channels (more than 256 channels), we can group neighboring channels or symmetric channels and tokenize them together, which could greatly shrink the final "sentence" length. However, additional adjustments are not needed in this paper and further discussion is beyond the scope of this paper.

