# OpenReview forum: "BIOT: Biosignal Transformer for Cross-data Learning in the Wild"
_NeurIPS.cc/2023/Conference — NeurIPS 2023 poster_

### Official Review · Reviewer_ykvi · 2023-07-03

**Soundness:** 3 good
**Presentation:** 3 good
**Contribution:** 3 good
**Rating:** 5
**Confidence:** 4

**Summary:**

This paper proposes a model called Biosignal Transformer (BIOT) that allows cross-dataset learning with different number channels, sequence lengths, and missing values. BIOT consists of a tokenization module that transforms multi-channel signals into a sequence (“sentence”), and a Linear Transformer Encoder to learn latent representations from the tokenized sequence. BIOT can be used for a wide variety of tasks, including self-supervised pre-training, supervised training (without vs with missing values), and supervised pre-training. Experiments suggest that BIOT outperforms existing methods on various biosignal classification tasks.

**Strengths:**

1. While Transformers have been widely used for modeling biosignals, there is some originality in the design of the token embeddings (segment embedding, channel embedding, and positional embedding).
2. Overall the methods are technically sound and easy to understand.
3. BIOT tackles the following challenges in modeling biosignals across datasets: mismatched channels, variable lengths, and missing values.

**Weaknesses:**

1. My major concerns about the method are: 1) its scalability to biosignals with long sequences and large number of channels and 2) spatial dependencies among channels are not appropriately modeled (e.g., graph structure of brain regions is not captured). These limitations should be investigated and discussed.
2. More details about token embeddings are needed.

**Questions:**

1. As mentioned above, I’m concerned about the scalability of BIOT to biosignals with long sequence lengths and large number of channels (which is the case for many real-world data). The datasets chosen have relatively short sequences and small number of channels. It would be great to see some experiments on the scalability. And its potential limitation should be discussed.
2. Many types of biosignals have graph structures (e.g., brain signals like EEGs). Flattening the channels into a sequence may be suboptimal for capturing such data structures. This limitation should be discussed.
3. Please provide more details about token embeddings. For example, how is the energy vector computed? What’s the dimensionality of the embedding table for channel embedding? What are the sinusoidal and cosine functions for positional embedding? If they follow previous studies, please provide citations.
4. More ablation studies are needed to show the impact of each component in BIOT. In particular, ablations for 1) normalization in the biosignal tokenization module and 2) each of the token embedding.

**Limitations:**

Some limitations are discussed in the paper. As mentioned above, please include additional discussion on the following limitations: 1) scalability to long sequences and large number of channels and 2) suboptimal for graph-structured data.

---

> ### Author Rebuttal · Authors · 2023-08-10
>
> # Response to reviewer ykvi
> ------
> ###### We thank the reviewer for the constructive feedback. We have uploaded a revision and used blue to mark the new changes. Our detailed responses are as follows.
>
> **Q1: My concern is on the scalability to biosignals with long sequences and large number of channels. It would be great to see some experiments on the scalability.**
>
> Thanks for bringing this point. The model can already handle long recordings and multiple channels in the following ways: (i) BIOT uses a linear complexity Transformer, so that the complexity scales linearly with sample lengths and channel sizes; (ii) Users can also remove the token overlaps and enlarge the token sizes to reduce the token numbers (i.e., "sentence" length).
>
> The BIOT model can better handle long recordings and multiple channels with minor adjustments. For long recordings, we could segment the recordings into 10-30s sessions, apply our BIOT on each session, and finally use a top-level LSTM or Transformer to learn sequence embedding. For multiple channels (more than 256 channels), we can group neighboring channels or symmetric channels and tokenize them together, which will greatly shrink the final "sentence" length. During the rebuttal, we added the scalability discussion in Appendix C.7.
>
> To provide more insights on the scalability of our model, we have provided a comparsion between linear transformer and original transformer (in terms of performance and runnning time) with different sample size inputs in Appendix C.6.
>
> - On the CHM-MIT dataset
> | Method                                 | Balanced Acc.        | AUC-PR              | AUROC              | Time per epoch       |
> |---|----------------------|---------------------|--------------------|----------------------|
> | BIOT (with linear Transformer)      | 0.6640 ± 0.0037      | 0.2573 ± 0.0088     | 0.8646 ± 0.0030    | 55.5780 ± 0.4229     |
> | BIOT (with naive Transformer)       | 0.6669 ± 0.0112      | 0.2493 ± 0.0088     | 0.8637 ± 0.0030    | 62.2877 ± 0.3629     |
>
> - On the IIIC Seizure dataset
> | Method                                | Balanced Acc.        | Cohen’s Kappa        | Weighted F1          | Time per epoch         |
> |---------------------------------------|----------------------|----------------------|----------------------|------------------------|
> | BIOT (with linear Transformer)     | 0.5762 ± 0.0034      | 0.4932 ± 0.0046      | 0.5773 ± 0.0031      | 25.4500 ± 4.0835       |
> | BIOT (with naive Transformer)      | 0.5810 ± 0.0029      | 0.4994 ± 0.0030      | 0.5822 ± 0.0044      | 28.2942 ± 4.0835       |
>
> - On the TUAB dataset
> | Method                                | Balanced Acc.        | AUC-PR               | AUROC               | Time per epoch        |
> |---------------------------------------|----------------------|----------------------|---------------------|-----------------------|
> | BIOT (with linear Transformer)     | 0.7925 ± 0.0035      | 0.8707 ± 0.0087      | 0.8691 ± 0.0033     | 25.2788 ± 0.1200      |
> | BIOT (with naive Transformer)      | 0.7902 ± 0.0033      | 0.8673 ± 0.0068      | 0.8652 ± 0.0052     | 27.3423 ± 0.2174      |
>
> - On the TUEV dataset
> | Method                               | Balanced Acc.        | Cohen’s Kappa        | Weighted F1          | Time per epoch       |
> |--------------------------------------|----------------------|----------------------|----------------------|----------------------|
> | BIOT (with linear Transformer)    | 0.4682 ± 0.0125      | 0.4482 ± 0.0285      | 0.7085 ± 0.0184      | 8.1812 ± 0.1228      |
> | BIOT (with naive Transformer)     | 0.4693 ± 0.0204      | 0.4512 ± 0.0317      | 0.7104 ± 0.0224      | 8.5493 ± 0.1814      |
>
> **Q2: Spatial dependencies among channels are not appropriately modeled (e.g., graph structure of brain regions is not captured). Flattening the channels into a sequence may be suboptimal for capturing such data structures.**
>
> Thanks for your question. The current model does not capture the graph structures of different channels (which is not the paper focus). However, whether or not to capture the spatial graphical information seems orthogonal to our BIOT model design. Our model can easily adopt designs from Transformer-based models if they have captured the channel graph structures. BIOT model can also add channel graph structures in other ways. For example, in addition to the segment/channel/positional embedding, our model can add a spatial embedding to the token embedding by utilizing channel graph representation learning. We have added the discussion in Appendix C.7.
>
> **Q3: How is the energy vector computed?**
>
> Thanks. We take FFT of each segment (e.g., 1s) and the results of FFT is the energy vector, where each entry is the energy for one frequency band.
>
> **Q4: What’s the dimensionality of the embedding table for channel embedding?**
>
> Thanks. The embedding table for channel embedding is a matrix, the row size is the channel number, and the col size is the embedding dimension.
>
> **Q5: What are the sinusoidal and cosine functions for positional embedding? If they follow previous studies, please provide citations.**
>
> Sure, thanks for reminding us. We have added it (Line 133) during the rebuttal. The same sinusoidal and cosine functions are used in the original transformer paper: Attention is All Your Needed.
>
> **Q6: More ablation studies are needed to show the impact of each component in BIOT. In particular, ablations for (1) normalization in the biosignal tokenization module and (2) each of the token embedding.**
>
> Sure, we have provided that in Appendix C.9. Different token embeddings are all important and are additively useful since they capture different information from the model: signal, frequency, and channel/spatial information. More details can refer to Appendix C.9.
>
> ---
> Thanks again for your time and valuable questions. We hope our new results and explanations can clear your concerns. We are happy to explain any other component of the model.

---

> > ### Comment · Reviewer_ykvi · 2023-08-13
> >
> > Thank you for your replies.
> >
> > Re: Q1, segmenting into short sessions may lose temporal correlations beyond 30s, which could be suboptimal for signals that have inherently long-range temporal correlations. Also, grouping neighboring channels may lose spatial information. Therefore, it would be more desirable for BIOT to be able to scale to long sequences and large number of channels without segmenting into shorter time windows or grouping channels.

---

> > > ### Author Response · Authors · 2023-08-13
> > >
> > > Thanks for your prompt replies. **We want to kindly provide further comments:**
> > >
> > > 1. The current BIOT model uses linear complexity transformer blocks, which scales linearly with recording lengths and channel numbers.
> > > 2. Segmenting into short sessions and then applying top-level LSTM/Transformer on session embeddings can potentially capture correlations beyond 30s (like a special version of Pyraformer [2]).
> > > 3. All designs that "can help vanilla transformer handle long sequences, such as [1][2][3][4]" can be used in our setting to help BIOT handle long recordings and multiple channels. The EEG/ECG samples in our paper are not long and do not have many channels, so we did not leverage these architectures in BIOT, while adding them is fairly straightforward.
> > >
> > > [1] Zaheer, Manzil, Guru Guruganesh, Kumar Avinava Dubey, Joshua Ainslie, Chris Alberti, Santiago Ontanon, Philip Pham et al. "Big bird: Transformers for longer sequences." Advances in neural information processing systems 33 (2020): 17283-17297.
> > >
> > > [2] Liu, Shizhan, Hang Yu, Cong Liao, Jianguo Li, Weiyao Lin, Alex X. Liu, and Schahram Dustdar. "Pyraformer: Low-complexity pyramidal attention for long-range time series modeling and forecasting." In International conference on learning representations. 2021.
> > >
> > > [3] Beltagy, Iz, Matthew E. Peters, and Arman Cohan. "Longformer: The long-document transformer." arXiv preprint arXiv:2004.05150 (2020).
> > >
> > > [4] Xiong, Yunyang, Zhanpeng Zeng, Rudrasis Chakraborty, Mingxing Tan, Glenn Fung, Yin Li, and Vikas Singh. "Nyströmformer: A nyström-based algorithm for approximating self-attention." In Proceedings of the AAAI Conference on Artificial Intelligence, vol. 35, no. 16, pp. 14138-14148. 2021.

---

### Official Review · Reviewer_XY8p · 2023-07-04

**Soundness:** 2 fair
**Presentation:** 1 poor
**Contribution:** 2 fair
**Rating:** 5
**Confidence:** 4

**Summary:**

Biological signals are crucial for clinical applications, but current models are specialized for specific settings such as sampling rate and duration. The authors propose a pre-trained model that enables cross-data training that addresses differences in sensor settings such as mismatched channels, variable sample lengths and sampling frequencies, and missing values. Compared to other baselines, their pretrained model improved balanced accuracy up to 7%.

**Strengths:**

•	The paper considers many different situations of biosignal applications: supervised learning with regularly-formatted data, supervised learning with irregularly data, unsupervised learning, pre-training on other datasets.
•	The authors apply many datasets in experiments, making the results convincing.


**Weaknesses:**

•	The proposed method lacks technical innovation. There isn’t significant technical contribution on top of Transformer or Vision-transformer (ViT). The majority of the paper is applying transformer to different application situations, but lacks the special model design for biosignals and its heterogeneity. While well suited for an application/health informatics venue the novelty does not extend sufficiently to NeurIPS levels.
•	The paper does not have a strong motivation or clearly declare it. From the abstract, the work (pre-trainable foundational biosignal models) is motivated by the success of LLM, however, chasing a hot topic should not be the motivation of a solid research work. Instead, it’s more important to discuss how LLM and transformer can be related to biosignal data.
•	The authors mention that mismatched channels, variable sample lengths, and prevalent missing values are unique challenges associated with biosignals, however, do not further explain why and how are they challenges.
•	Even though Transformer is widely applied in different application, recurrent model is still an important baseline model especially for time-series data.
•	I’m not sure if it’s appropriate to say “biosignals are more complicated” (Line 67). Other data modalities/applications may have their own challenges too.
•	The paper is not well written, for example, many paragraphs in introduction do not have connections to the previous or later one.
•	More than one tables are too big and out of the range.


**Questions:**

•	What are the limitations of using language modeling algorithms for biosignal tasks and considering biosignals as "sentences"?
•	Why would this tokenization scheme not work across domains (e.g. using Pre Trained EEG on ECG) as part of the unified biosignal model?
•	What are the hyperparameter tuning range?

**Limitations:**

•	The authors mention that performance for other types of tasks were not compared, but they also don’t discuss whether the tokenization scheme would affect the performance of those tasks differently.

---

> ### Author Rebuttal · Authors · 2023-08-10
>
> # Response to reviewer XY8p
> ----
> ##### We thank the reviewer for the helpful feedback. We have uploaded a revision with the changes marked as blue. Our detailed responses are as follows:
>
> **Q1: What are the limitations of using language modeling algorithms for biosignal tasks and considering biosignals as "sentences"?**
>
> As we mentioned in the paper, many biosignals could have mismatched channels, missing values, or variable lengths across different datasets, and thus directly applying language modeling algorithms or other sequence models on these settings are suboptimal or infeasible. *This is our motivation for developing the BIOT encoder architecture.*
>
> **Q2: There isn’t significant technical contribution on top of Transformer or Vision-transformer (ViT). The majority of the paper is applying transformer to different application situations, but lacks the special model design for biosignals and its heterogeneity.**
>
> Thanks, this is a misunderstanding. Our contribution is that common Transformer models or ViT models cannot handle irregular biosignals (with mismatched channels, missing values, or variable lengths), while in this paper, we transform different signals into unified formats and thus enable the joint learning and generalizing to downstream applications across different datasets. Our model is designed specifically for biosignals: the tokenization transformation is designed based on multi-channel structures of biosignals, and the segmentation step also follows the frequency representations of biosignals.
>
> **Q3: The paper does not have a strong motivation or clearly declare it. From the abstract, the work (pre-trainable foundational biosignal models) is motivated by the success of LLM, however, chasing a hot topic should not be the motivation of a solid research work. Instead, it’s more important to discuss how LLM and transformer can be related to biosignal data.**
>
> Thanks, to reduce the confusion, we have rephrased our Abstract during the rebuttal. In the original Abstract and Introduction, we have mentioned that this paper is motivated by the fact that many previous biosignal learning works cannot apply/generalize to other settings. This paper proposes a powerful encoder architecture that is able to incorporate different datasets in training and is flexible in generalization since it can handle mismatched channels, variable lengths, and missing values. The connection to previous text modeling (LLM) work is that Transformers can handle variable lengths, and thus we are motivated to design a tokenization trick to transform biosignals into variable-length "sentences".
>
> **Q4: The authors mention that mismatched channels, variable sample lengths, and prevalent missing values are unique challenges associated with biosignals, however, do not further explain why and how are they challenges. I’m not sure if it’s appropriate to say “biosignals are more complicated” (Line 67). Other data modalities/applications may have their own challenges too.**
>
> Thanks. The original Introduction explained why and how these three are challenging (Line 44 - 53). In additional, there are already previous works discussing how to transform images (ViT), audios (Wave2Vec), and natural languages into a sentence, and this work proposes a way to transform multichannel biosignals into a sentence. We mention it is more challenging in the sense that previously there is no relevant study on biosignal transformation. Based on the reviewer's comments, we further rephrased this part (Line 67-68) during the rebuttal.
>
> **Q5: Even though Transformer is widely applied in different application, recurrent model is still an important baseline model especially for time-series data.**
>
> Yes, two baselines included in this paper CNN-Transformer (Peh et al., 2022) and FFCL (Li et al., 2022) are based on LSTM or contain a LSTM component, and they are inferior to our proposed model under various evaluations.
>
> **Q6: The paper is not well written, for example, many paragraphs in introduction do not have connections to the previous or later one. More than one tables are too big and out of the range.**
>
> Thanks, we have rephrased the relevant parts and adjusted the size of the tables.
>
> **Q7: Why would this tokenization scheme not work across domains (e.g. using Pre Trained EEG on ECG) as part of the unified biosignal model?**
>
> Thanks. We did not try the tokenization scheme and do not know whether it works or not across domains. The current paper did not train EEG and ECG or other signal types together since different types are semantically different, such as in sampling frequencies, signal amplitudes, and time series microstructures. Also the corpus used for pre-training might also need to align with or remotely connected to the downstream tasks (for example, currently it is unclear how ECG signals can help EEG learning tasks). But, unifying different signals can be a promising direction as future work.
>
> **Q8: What are the hyperparameter tuning range?**
>
> Thanks. The initial Appendix C.3 has reported our special hyperparameters ablation studies, including frequency choices, overlapping lengths, and token lengths. This ablation study can provide insights on how to select our newly introduced hyperparameters. Other hyperparameters, such as learning rate, batch size, L2 regularizers, are selected following common procedures based on the validation set.
>
> **Q9: The authors mention that performance for other types of tasks were not compared, but they also don’t discuss whether the tokenization scheme would affect the performance of those tasks differently.**
>
> Sorry, we kindly ask the reviewer to clarify the question. We would be happy to address your concerns if more details are provided (such as Line number).
>
> ----
> Thanks again for your comments. Hope our rebuttal has addressed all your concerns.

---

> > ### Comment · Reviewer_XY8p · 2023-08-15
> > **Thank you for your thorough response**
> >
> > Thanks for the diligent response and clarification.
> >
> > I still have a few outstanding questions that might help you in your final paper.
> >
> > 1. the missingness of the signal-type times-series data is often long-term instead of one or very few time windows, like from disconnection to a device, and thus I have concern about the authors’ problem setup - is the data missing problem really going to be in as in Figure 2, to apply supervised learning with missing data or unsupervised learning with contrastive loss? I would suggest the authors to have more description of the data missing problem in biosignal data, maybe provide some examples.
> >
> > 2. There isn’t enough introduction about the baseline models. Do they also have other dataset involved in training? If not then it may not be a fair comparison to the BIOT at least the pre-trained BIOT.

---

> > > ### Author Response · Authors · 2023-08-16
> > >
> > > Thanks for your questions. We can briefly clarify them below.
> > >
> > > 1. **For the problem setting**. Figure 2 is just for illustration purpose (it does not necessarily imply any real application). In the real world, missingness in signal data could be long-term due to disconnection of device. It could also be multiple short-term windows due to the connection is unstable and occasionally getting disconnected. Also, handling missing signals is just one application in the paper, and the model can also handle other scenarios, such as misaligned channels. **The most important message here is that our framework is generic enough to handle all different scenarios of signal missings (long-term or short-term), channel misalignments, and different signal lengths.** In Section 3.3, we verified the strengths of our BIOT model on both scenarios.
> > >
> > > 2. **More information on baseline models.** In Appendix B.2, we do have extra information for baseline models (such as how they are implemented). In the experimental Table 2 and Table 3, the baseline models and our Vanilla BIOT model only uses a single dataset ,while the pre-trained model is trained on other datasets first and then finetuned on this dataset. We mainly want to show: (i) Vanilla BIOT works better than the baseline models consistently; (ii) With the BIOT framework, we can transfer knowledges from other datasets (with different signal length and channels) and improve the prediction performance on this dataset, which previous models cannot do. Here, we do not intend to compare the baseline with the pre-trained BIOT.
> > >
> > > We hope this can address your additional concerns. We thank the reviewer again and will further improve the final paper based on your valuable comments.

---

### Official Review · Reviewer_vQ39 · 2023-07-05

**Soundness:** 3 good
**Presentation:** 3 good
**Contribution:** 3 good
**Rating:** 7
**Confidence:** 4

**Summary:**

The authors present a general Transformer-based pipeline for learning on biosignals such as EEG, ECG and human activity sensor data. The proposed approach relies on a tokenization scheme that includes temporal and channel position information and a spectral representation of a segment of a single-channel time series. This allows learning and inference on examples with missing channels and/or time points. The impact of contrastive pretraining is evaluated on multiple modality-specific downstream tasks (EEG seizure detection, ECG arrythmia prediction, etc.), as well as the impact of missing channels/segments and of supervised pretraining. Results compare favorably to existing baselines.

**Strengths:**

Originality: The paper proposes an original approach to feed multivariate time series data into a Transformer through a bespoke tokenization scheme. The combination with a contrastive unsupervised pretraining task and the applicability to multiple different modalities is also novel.

Quality: The paper is technically sound, with results presented on multiple downstream tasks and different modalities. There are multiple analyses supporting the choice of hyperparameters.

Clarity: The paper is overall clear and well organized. The provided code is very well organized and clearly formatted.

Significance: the developed methodology is likely to be reused and could yield pretrained "backbone" models that can be finetuned on different tasks. Reported improvements on different baseline tasks also suggest this approach can outperform SOTA models without pretraining. Finally, the model's ability to naturally work despite missing channels and segments is compelling.

**Weaknesses:**

Quality: some of the core choices behind the methodology could be better supported/justified. For instance: the use of a linear complexity attention module (see question 1), the choice of a spectral representation for the segment embedding (Q2) and the use of a contrastive loss with channel/segment dropout augmentation (instead of a different unsupervised task, e.g. masked autoencoding) for unsupervised pretraining.

**Questions:**

1. The use of an attention implementation with linear complexity is compelling, especially if the tokenization scheme yields long sequences. Can you give a sense of this length for the experiments of Table 2 and 3? Given the results are already above the baselines choosing linear attention might be a good tradeoff, but it would be interesting to present a comparison with a vanilla Transformer block for at least one of the experiments (both in terms of performance and of running time like in Table 7).
2. What is the impact of using a spectral (FFT) representation rather than the time series segment itself as the input to the segment embedding FCN? The phase information that is discarded might actually be useful for some downstream tasks, e.g. when fine microstructure is relevant to a task or there are events/stimuli in a stimulation protocol.
3. At line 212, it is said that "the first three datasets are used entirely for unsupervised pre-training"; there are no splitting information available for these datasets in the Appendix (and the code only defines a training dataloader) and so I understand that the hyperparameters were selected based on downstream task validation performance only. It would be interesting to see how pretext task performance relates to downstream task performance, and whether it is similar across modalities (however this is purely curiosity from my part, I don't think such an analysis is necessary for this submission).
4. At line 184, T=0.2, but in the appendix (line 565) it is said to be T=2.
5. The use of the word "montages" in an EEG context in Table 1 and in Appendix B is confusing. Typically "montages" refer to a fixed set of channel positions or derivations. Therefore, line 483 should instead read something like "First, the 16 derivations ...".

**Limitations:**

Yes.

---

> ### Author Rebuttal · Authors · 2023-08-10
>
> # Response to reviewer vQ39
> -------
> ##### We thank the reviewer for your appreciation and constructive comments. We have uploaded a revision and used blue to mark the new changes.
> **Q1: Can you give a sense of this length for the experiments of Table 2 and 3?**
>
> Sure, it can be calculated from Table 1. For example, for the SHHS dataset, we use 0.5s as token overlap and 1s as the token length. Then, a data sample with 30s duration will have (30 * 2 - 1) * 2 channels = 118 tokens. For PREST, we use 0.5s as token overlap and 1s as token length. Then, a data sample with 10s duration will have (10 * 2 - 1) * 16 = 304 tokens. Sample durations are all given from the data resources.
>
> **Q2: It would be interesting to present a comparison with a vanilla Transformer block for at least one of the experiments (both in terms of performance and of running time like in Table 7).**
>
> Sure, we provided the comparison in Appendix C.6 (and copyed the results below). Generally, we find that the BIOT model with vanilla Transformer costs a bit more time while the performance of using linear complexity Transformer and vanilla Transformer are very similar. The reason might be that biosignals, such as EEG, are known to have *low-rank structure naturally*, and thus the low-rank approximation of the quadratic attention (in linear complexity Transformer) will not lead to a noticeable performance drop.
>
> - On the CHM-MIT dataset
> | Method   | Balanced Acc.    | AUC-PR  | AUROC | Time / epoch (s) |
> |--|--|--|---|--|
> | BIOT (with linear Transformer)    | 0.6640   | 0.2573     | 0.8646   | 55.5780 |
> | BIOT (with naive Transformer)    | 0.6669     | 0.2493      | 0.8637   | 62.2877   |
>
> - On the IIIC Seizure dataset
> | Method    | Balanced Acc.  | Cohen’s Kappa | Weighted F1  | Time / epoch (s)   |
> |--|--|--|---|--|
> | BIOT (with linear Transformer)     | 0.5762  | 0.4932    | 0.5773       | 25.4500    |
> | BIOT (with naive Transformer)      | 0.5810     | 0.4994   | 0.5822      | 28.2942  |
>
> - On the TUAB dataset
> | Method    | Balanced Acc.     | AUC-PR    | AUROC    | Time / epoch  (s)  |
> |--|--|--|---|--|
> | BIOT (with linear Transformer)   | 0.7925  | 0.8707  | 0.8691      | 25.2788       |
> | BIOT (with naive Transformer)      | 0.7902   | 0.8673  | 0.8652      | 27.3423       |
>
> - On the TUEV dataset
> | Method | Balanced Acc. | Cohen’s Kappa  | Weighted F1  | Time / epoch (s)  |
> |--|--|--|---|--|
> | BIOT (with linear Transformer)    | 0.4682 | 0.4482  | 0.7085  | 8.1812   |
> | BIOT (with naive Transformer)     | 0.4693  | 0.4512  | 0.7104  | 8.5493  |
>
> **Q3: What is the impact of using a spectral (FFT) representation rather than the time series segment itself as the input to the segment embedding FCN? The phase information that is discarded might actually be useful for some downstream tasks.**
>
> Thanks, this is a valuable question. In our experiments, we use the FFT representation since relevant tasks in the paper would benefit more from the spectral domain than the raw signal microstructures. However, if the phase information is important in other applications, our BIOT can be flexible to use raw time series as inputs.
>
> **Q4: It would be interesting to see how pretext task performance relates to downstream task performance, and whether it is similar across modalities (however this is purely curiosity from my part, I don't think such an analysis is necessary for this submission).**
>
> Thanks for your question, and this would lead to interesting discussions. We need to consider two aspects when analyzing the generalization performance. **First**, whether the pre-trained data is semantically (in terms of the downstream task) close to the downstream task data. **Second**, whether the pretraining task itself is optimized well (this is the reviewer's question). Intuitively, only when the pretraining data is close to the downstream data and the pretraining task is optimized well, then the downstream performance can be greatly improved.
>
> Actually, Figure 4 and Figure 5 can verify the first aspect, given that the second aspect is satisfied (the supervised pretraining models are all optimized well). From Figure 4 and Figure 5, we can see that pretraining on more similar datasets will generate better gain in downstream finetuning. For the second aspect, we conduct experiments in Appendix C.8 by loading the checkpoints of pretrained model from different epochs, which shows that the generalization performance will be better if the pretrained model is optimized better. The conclusion is consistent on both EEG and ECG datasets.
>
>
> - Load the checkpoints of pretrained BIOT on PREST at different epochs (1, 2, 3, 4, 5, 8, 10, 20) and finetuning on the IIIC Seizure dataset.
> | Checkpoint Epoch | 1 |2| 3| 4| 5| 8 | 10 | 20 |
> |----| ---- | ---- | ---- | ---- | ---- | ---- | ---- | ---- |
> | loss in pretraining | 17.32 | 6.19 | 1.94|  0.93 | 0.63 | 0.42 | 0.33 | 0.31 |
> | Weighted F1 when finetuning on IIIC Seizure | 0.5773 | 0.5815 |  0.5804 | 0.5822 | 0.5813 | 0.5835 | 0.5825 | 0.5828 |
>
> - Load the checkpoints of pretrained BIOT on Cardiology-12 at different epochs (1, 2, 3, 4, 5, 8, 10, 20) and finetuning on the PTB-XL dataset.
> | Checkpoint Epoch |  1 |2| 3| 4| 5| 8 | 10 | 20 |
> |--| -- | ---- | ---- | ---- | ---- | ---- | ---- | ---- |
> | loss in pretraining | 27.23 |  9.49 | 2.59 | 1.57 | 1.13 | 1.23| 1.24 | 1.01 |
> | Weighted F1 when finetuning on IIIC Seizure | 0.7493 | 0.7525 | 0.7631 | 0.7689 | 0.7645 | 0.7684 | 0.7652 | 0.7671 |
>
>
> **Q5: At line 184, T=0.2, but in the appendix (line 565) it is said to be T=2.**
>
> Thanks for finding this typo. T=0.2 is correct.
>
> **Q6: The use of the word "montages" in an EEG context in Table 1 and in Appendix B is confusing. Typically "montages" refer to a fixed set of channel positions or derivations. Therefore, line 483 should instead read something like "First, the 16 derivations ...".**
>
> Thanks for your suggestion. We have updated these places (Line 482, Line 500, Line 506, Line 512) in the paper.

---

> > ### Comment · Reviewer_vQ39 · 2023-08-15
> >
> > Thank you to the authors for providing additional results and clarifications.

---

### Official Review · Reviewer_ve81 · 2023-07-11

**Soundness:** 3 good
**Presentation:** 3 good
**Contribution:** 3 good
**Rating:** 7
**Confidence:** 4

**Summary:**

The paper presents a Biosignal Transformer (BIOT) model that can be pre-trained from multiple data sources and fine-tuned on different downstream biosignal tasks. The model tokenizes diverse biosignals into unified “biosignal sentences” and adds channel embeddings and relative position embeddings to preserve spatio-temporal features. The BIOT model is versatile and applicable to various biosignal learning settings across different datasets. Comprehensive evaluations on EEG, ECG, and human activity sensory signals demonstrate that BIOT outperforms robust baselines in common settings and facilitates learning across multiple datasets with different formats.



**Strengths:**

The paper presents a novel approach to handling diverse biosignals by tokenizing them into unified “biosignal sentences”. The proposed BIOT model is versatile and applicable to various biosignal learning settings across different datasets. The comprehensive evaluations on EEG, ECG, and human activity sensory signals demonstrate the robustness of the proposed model.



**Weaknesses:**

The paper does not provide motivation for why the authors picked these specific datasets. There is no discussion about large datasets like MIMIC-IV that include ECG data. The selected baselines do not include some reference models like SimCLR or MAEs.



**Questions:**

Why were these specific datasets chosen for evaluation?
How would the proposed model perform on large datasets like MIMIC-IV?
How would the proposed model compare to reference models like SimCLR or MAEs?


**Limitations:**

The most significant limitation of this paper is the experimental setup. The selected baselines do not include some reference models like SimCLR or MAEs. There are no scaling experiments which seems to be the most important finding in GPT-style models. If we want to create the equivalent of a GPT for biosignals, we need to source and combine probably hundreds of similar datasets and think hard about the proportion of each modality (EEG, movement, ECG etc). However, the authors left this for future work.

---

> ### Author Rebuttal · Authors · 2023-08-10
>
> # Response to reviewer ve81
>
>
> ###### We thank the reviewer for the constructive feedback. We have uploaded a revision and used blue to mark the new changes. Our detailed responses are as follows.
> -----
> **Q1: The paper does not provide motivation for why the authors picked these special datasets. There is no discussion about large datasets like MIMIC-IV that include ECG data.**
>
> Thanks. The datasets used in the paper are all large and widely used EEG/ECG datasets. Previous works (such as [1][2][3][4][5]) also use these datasets. Some (such as SHHS, CHB-MIT, Cardiology, PTB-XL) are from the open PhysioNet repository, some (such as TUEV, TUAB) are from Temple University open EEG corpus, and some (such as PREST, IIIC-Seizure) are proprietary datasets. In terms of the dataset size, SHHS (portion 1) has ~185 GB, PREST has ~320 GB, Cardiology has ~77 GB, TUAB has ~159 GB, TUEV has ~37 GB, etc. Also, the recommended MIMIC-IV-ECG dataset is officially under embargo (please check the MIMIC-IV-ECG website), and we are unable to provide additional results.
>
> [1] McCallan, Niamh, Scot Davidson, Kok Yew Ng, Pardis Biglarbeigi, Dewar Finlay, Boon Leong Lan, and James McLaughlin. "Epileptic multi-seizure type classification using electroencephalogram signals from the Temple University Hospital Seizure Corpus: A review." Expert Systems with Applications (2023): 121040.
>
> [2] Yang, Chaoqi, M. Brandon Westover, and Jimeng Sun. "Manydg: Many-domain generalization for healthcare applications." ICLR (2023).
>
> [3] Prasanna, J., M. S. P. Subathra, Mazin Abed Mohammed, Robertas Damaševičius, Nanjappan Jothiraj Sairamya, and S. Thomas George. "Automated epileptic seizure detection in pediatric subjects of CHB-MIT EEG database—a survey." Journal of Personalized Medicine 11, no. 10 (2021): 1028.
>
> [4] Sridhar, Niranjan, Ali Shoeb, Philip Stephens, Alaa Kharbouch, David Ben Shimol, Joshua Burkart, Atiyeh Ghoreyshi, and Lance Myers. "Deep learning for automated sleep staging using instantaneous heart rate." NPJ digital medicine 3, no. 1 (2020): 106.
>
> [5] Biswal, Siddharth, Haoqi Sun, Balaji Goparaju, M. Brandon Westover, Jimeng Sun, and Matt T. Bianchi. "Expert-level sleep scoring with deep neural networks." Journal of the American Medical Informatics Association 25, no. 12 (2018): 1643-1650.
>
> **Q2: The selected baselines do not include some reference models like SimCLR or MAEs. How would the proposed model compare to these reference models.**
>
> Thanks, this is a misunderstanding. Our paper aims to provide a *new biosignal encoder architecture* that can handle different signal formats (with missing, variable lengths, and mismatched channels) at once. Our idea is orthogonal to the design of SimCLR or MAE, and instead, BIOT can be a backbone for them under the self-supervised setting. Note that our BIOT encoder combined with a final prediction layer can be a supervised model and combined with SimCLR/MAE/NCE can be an unsupervised model.
>
> Per the reviewer's request, we use BIOT as the backbone and try SimCLR and MAE (conservative and aggressive version with different data augmentation rates) in the self-supervised setting (to replace the current NCE loss in Equation 4). We use TUEV and TUAB as the datasets, show the new results in Appendix C.5, and copy them here. Detailed result analysis can be found in Appendix C.5. *Note that we want to kindly remind the reviewer that this paper proposes a new biosignal encoder architecture but not a new self-supervised learning framework*.
>
> | Model | (TUAB) Balanced Acc. | (TUAB) AUC-PR | (TUAB) AUROC | (TUEV) Balanced Acc. | (TUEV) Coken’s Kappa | (TUEV) Weighted F1 |
> |------|---|---------------------|---|---|----|---------------------|
> | BIOT + NCE (PREST)                     | 0.7907 ± 0.0050     | 0.8752 ± 0.0051     | 0.8730 ± 0.0021     | 0.5207 ± 0.0285     | 0.4932 ± 0.0301     | 0.7381 ± 0.0169     |
> | BIOT + NCE (PREST+SHHS)                | 0.8019 ± 0.0021     | 0.8749 ± 0.0054     | 0.8739 ± 0.0019     | 0.5149 ± 0.0292     | 0.4841 ± 0.0309     | 0.7322 ± 0.0196     |
> | BIOT + SimCLR (PREST)                  | 0.7894 ± 0.0072     | 0.8681 ± 0.0053     | 0.8715 ± 0.0074     | 0.5113 ± 0.0331     | 0.4862 ± 0.0209     | 0.7310 ± 0.0244     |
> | BIOT + SimCLR (PREST+SHHS)             | 0.7852 ± 0.0096     | 0.8655 ± 0.0014     | 0.8694 ± 0.0075     | 0.5147 ± 0.0311     | 0.4829 ± 0.0157     | 0.7280 ± 0.0348     |
> | BIOT + conservative MAE (PREST)        | 0.7393 ± 0.0087     | 0.8347 ± 0.0076     | 0.8259 ± 0.0081     | 0.4760 ± 0.0328     | 0.4393 ± 0.0420     | 0.6831 ± 0.0112     |
> | BIOT + conservative MAE (PREST+SHHS)   | 0.7410 ± 0.0098     | 0.8312 ± 0.0082     | 0.8262 ± 0.0061     | 0.4601 ± 0.0238     | 0.4349 ± 0.0182     | 0.6797 ± 0.0189     |
> | BIOT + aggressive MAE (PREST)          | 0.7679 ± 0.0045     | 0.8591 ± 0.0086     | 0.8412 ± 0.0068     | 0.4929 ± 0.0416     | 0.4728 ± 0.0364     | 0.7130 ± 0.0216     |
> | BIOT + aggressive MAE (PREST+SHHS)     | 0.7692 ± 0.0011     | 0.8525 ± 0.0079     | 0.8459 ± 0.0043     | 0.4970 ± 0.0409     | 0.4686 ± 0.0136     | 0.7086 ± 0.0195     |
>
>
>
> **Q3: There are no scaling experiments which seems to be the most important finding in GPT-style models. If we want to create the equivalent of a GPT for biosignals, we need to source and combine probably hundreds of similar datasets and think hard about the proportion of each modality (EEG, movement, ECG etc). However, the authors left this for future work.**
>
> Thanks. We have improved the Abstract and Introduction to clarify our contributions. As we mentioned earlier, *this paper barely provides a powerful encoder architecture that is able to incorporate different datasets in training and is flexible in generalization*. This work does not claim to provide a biosignal GPT (which requires huge time efforts and computing resources), and we leave it as a future work.
>
> ----
> Thanks again for the valuable comments. Hope our rebuttal has addressed all your concerns.

---

> > ### Comment · Reviewer_ve81 · 2023-08-17
> >
> > Thanks for the thorough response and for taking my suggestions into account. I am happy to increase my score to Accept now.

---

### Author Rebuttal · Authors · 2023-08-10

We thank all the reviewers for your time and constructive feedback. During the rebuttal, we have prepared a revision and used blue to mark the new changes.

---

### Decision · Program_Chairs · 2023-09-21

**Decision:**

Accept (poster)

**Comment:**

Thank you for your submission. Reviewers agreed that the experimental approach was well written (vQ39, ykvi), the introduction and motivation were weaker (XY8p), with some methodological detail missing (ykvi). The use of multiple datasets and evaluations was appreciated by all reviewers, although these datasets appeared arbitrarily chosen (ve81). The rebuttal is weak for this latter point; datasets being popular and large is not motivating. Given the wealth of biosignal data available, selection of datasets to evaluate specific aspects of the encoder structure is warranted (e.g. scaling to long sequences via Holter monitor data would address concerns of reviewer ykvi). Reviewers found the new encoder structure of interest (ve81, ykvi). There was limited justification of why mismatched channels, missing values, and variable lengths are the defining challenges of biosignals (XY8p), as opposed to other clear issues (non-stationarity, multiple scales, varying sampling frequencies, FFT vs. raw signal, etc). The missing data approximations are unrealistic (XY8p). There is an implication that this encoder could work for all biosignals (ECG, EEG, ABP, EMG, EOG, CVP, ICP, respirometry, actimetry, PPG, etc), the evaluation would be stronger with more biosignal modalities included as it heavily focuses on EEG currently. Scaling the model remains an unaddressed challenge (ykvi), as the rebuttal experiments focus on the subsequent components (linear transformer) rather than the embedding approach itself.

On balance the work seems a worthwhile contribution with room for improvement in many areas.